# RE-TRAC: REcursive TRAjectory Compression for Deep Search Agents

Jialiang Zhu[*][†][1]  Gongrui Zhang[*][†][1]  Xiaolong Ma[*][†][2]  Lin Xu[*][†][3]  Miaosen Zhang[†][1]  Ruiqi Yang[†][4]
Song Wang[†][5]  Kai Qiu[*][6]  Zhirong Wu[*][6]  Qi Dai[6]  Ruichun Ma[6]  Bei Liu[6]  Yifan Yang[6]  Chong Luo[6]
Zhengyuan Yang[6]  Linjie Li[6]  Lijuan Wang[6]  Weizhu Chen[6]  Xin Geng[1]  Baining Guo[1][6]

## Abstract

LLM-based deep research agents are largely built on the ReAct framework. This linear design makes it difficult to revisit earlier states, branch into alternative search directions, or maintain global awareness under long contexts, often leading to local optima, redundant exploration, and inefficient search. We propose Re-TRAC, an agentic framework that performs cross-trajectory exploration by generating a structured state representation after each trajectory to summarize evidence, uncertainties, failures, and future plans, and conditioning subsequent trajectories on this state representation. This enables iterative reflection and globally informed planning, reframing research as a progressive process. Empirical results show that Re-TRAC consistently outperforms ReAct by 15–23% on BrowseComp with frontier LLMs. For smaller models, we introduce Re-TRAC-aware supervised fine-tuning, achieving state-of-the-art performance at comparable scales. Notably, Re-TRAC shows a monotonic reduction in tool calls and token usage across rounds, indicating progressively targeted exploration driven by cross-trajectory reflection rather than redundant search. Code and models are available at `https://github.com/microsoft/InfoAgent`.

## 1. Introduction

Large language models (LLMs) have progressed from single-turn question answering to chain-of-thought reasoning (Wei et al., 2022), function calling (Schick et al., 2023), and complex multi-turn agentic applications (Anthropic, 2025). This evolution reflects a shift from passive response generation toward autonomous, goal-directed problem solving in open environments. A deep research agent (OpenAI, 2025a; Google, 2025), capable of autonomously searching the open web and gathering and analyzing information from thousands of web pages, represents the next frontier of information retrieval for general intelligence.

Most existing deep research agents are built upon the ReAct paradigm (Yao et al., 2023b), which interleaves large language model (LLM) reasoning steps with tool invocation, appending both into the model context in a linear sequential manner. In this work, we provide an in-depth analysis of the inherent limitations of ReAct-style linear reasoning workflows. Although LLM reasoning can be trained to support behaviors such as backtracking and self-reflection (Guo et al., 2025), this strictly linear agentic workflow is not well suited for open-ended tasks that require broad exploratory investigation. Revisiting earlier reasoning states and branching into alternative search trajectories remains challenging, particularly under long-context settings (e.g., 128K–256K tokens), where context management and credit assignment become increasingly difficult. Consequently, the ReAct framework is susceptible to issues such as local optima, redundant exploration, and inefficient search dynamics (Yao et al., 2023a).

To empower LLM-based agents with diverse exploration capabilities, we propose to explicitly guide agents toward search trajectories that have not been previously explored. This direction is motivated by two key observations. First, existing deep research models (even after extensive reinforcement learning post-training) exhibit substantially higher pass@k performance than pass@1. This gap indicates that repeated inference induces diverse reasoning trajectories, suggesting that model limitations often stem from insufficient exploration within a single trajectory rather than inadequate reasoning capacity. Second, prior work shows that LLMs are generally better at verifying candidate solutions than generating them from scratch (Weng et al., 2023; Singhi et al., 2025), motivating a search paradigm that emphasizes broad candidate generation followed by

---

[*]Equal Core Contributors †This work was done during the internship at MSRA [1]Southeast University [2]Waseda University [3]Tsinghua University [4]Brown University [5]Zhejiang University [6]Microsoft. Correspondence to: Baining Guo <307000167@seu.edu.cn>, Xin Geng <xgeng@seu.edu.cn>, Kai Qiu <Kai.Qiu@microsoft.com>, Zhirong Wu <Wu.Zhirong@microsoft.com>.

*Proceedings of the $43^{rd}$ International Conference on Machine Learning*, Seoul, South Korea. PMLR 306, 2026. Copyright 2026 by the author(s).

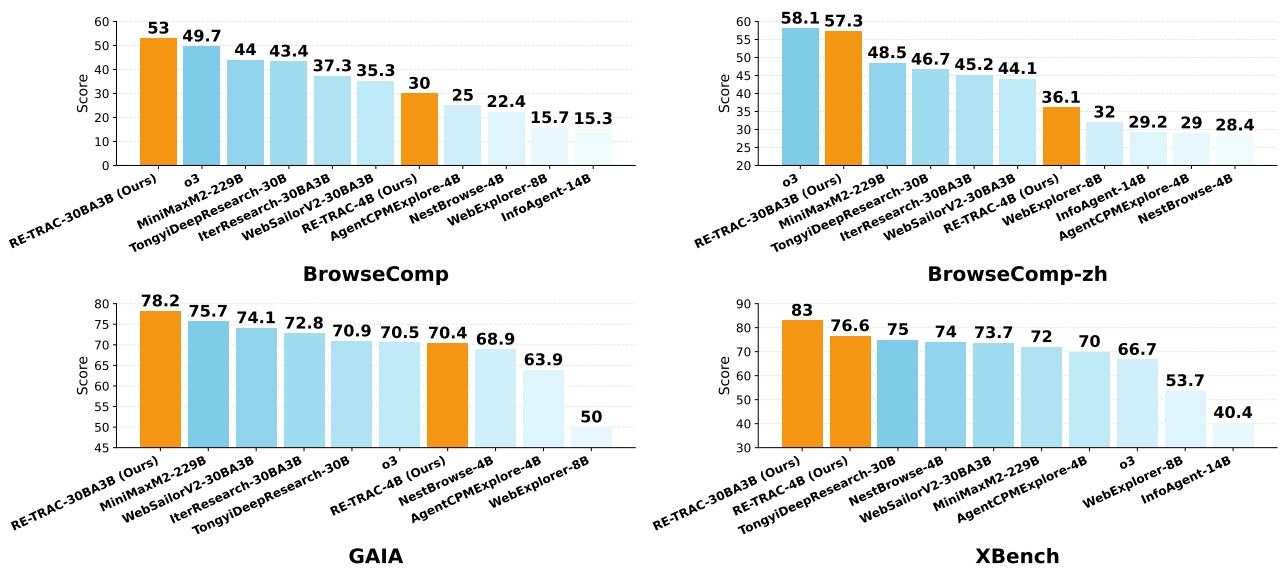

*Figure 1.* Comparison of RE-TRAC with state-of-the-art agentic models. Our 4B and 30B models surpass the performance of significantly larger, state-of-the-art models.

verification-driven selection.

We propose Re-TRAC, an agentic framework that recursively constructs structured state representation at the end of each trajectory and uses them as the prompting context for subsequent trajectories. Each state representation summarizes the evolving state of investigation along multiple dimensions, including accumulated evidence, unresolved uncertainties, identified failure modes, and a forward-looking research plan. Unlike multiple independent trajectories that operate in isolation, Re-TRAC enables iterative reflection, cross-trajectory knowledge consolidation, and globally informed planning. This design transforms exploration from a set of disconnected attempts into a progressively informed search process. Empirically, we observe that Re-TRAC agents issue fewer tool calls and consume fewer tokens with each successive round of research, indicating improved decision-making efficiency and more targeted information acquisition guided by prior experience.

Our experiments demonstrate that Re-TRAC achieves absolute gains of 15–23% over ReAct on the BrowseComp benchmark when applied with frontier LLMs. This inspires us to push the limits of abilities of small models via Re-TRAC. To unlock the benefits of Re-TRAC for smaller models, we develop a post-training recipe that constructs supervised fine-tuning (SFT) data consisting of trajectories explicitly conditioned on structured state representations. This training procedure teaches the model to ground its reasoning, planning, and tool use on structured cross-trajectory summaries rather than relying solely on immediate context. After fine-tuning, our 30B model achieves 53% accuracy on BrowseComp, while the 4B model reaches 30%, es-

tablishing state-of-the-art performance among models of comparable scale (see Figure 1).

## 2. Related Work

### 2.1. Deep Research Agents

The emergence of Deep Research Agents marks a transition from simple information retrieval to autonomous systems capable of long-horizon reasoning, strategic planning, and persistent tool utilization (OpenAI, 2025a; Google, 2025; xAI, 2025; Liu et al., 2025a; Perplexity, 2025; MiniMax, 2025; Zhang et al., 2025; Team et al., 2025b; Team, 2025a). Agents powered by proprietary models, such as OpenAI Deep Research (OpenAI, 2025a), Gemini Deep Research (Google, 2025), Claude (Antropic, 2025), Perplexity (Perplexity, 2025), and Grok (xAI, 2025), leverage large-scale training and deep tool integration to achieve high accuracy. In parallel, open-source models, including DeepSeek (Liu et al., 2025a), GLM (Team, 2025a), Kimi (Team et al., 2025a), MiniMax (MiniMax, 2025), and Tongyi Deep Research (Team et al., 2025b), have strengthened their long-horizon capabilities through specialized training on extensive agentic tasks. Additionally, works like InfoAgent (Zhang et al., 2025), WebSailor (Li et al., 2025c), and Deep-Dive (Lu et al., 2025) have explored foundational challenges such as data synthesis and search-oriented environment construction. Our work introduces a recursive experience compression mechanism to enhance the agent's ability to handle long-horizon tasks.

## 2.2. Agentic Context Management

The ability to manage context effectively is critical for agents performing long-horizon tasks. Recent research generally falls into two categories: intrinsic context optimization (Liu et al., 2025a; Team, 2025a) and external memory mechanisms for state maintenance (Chen et al., 2025a; Yu et al., 2025; Wu et al., 2025; Zhou et al., 2025b; Song et al., 2025). For the first category, many agentic LLMs, such as DeepSeek-V3.2 (Liu et al., 2025a), and GLM-4.7 (Team, 2025a), integrate context pruning directly into the agent's reasoning loop, focusing on compressing observation spaces and pruning redundant trajectory history. Parallel to context pruning, recent works have focused on leveraging external memory. IterResearch (Chen et al., 2025a) and MemAgent (Yu et al., 2025) utilize dynamic memory structures to reconstruct task status at each step, discarding generic history to simulate infinite horizons. Re-Sum (Wu et al., 2025) introduces a "summarize-and-reset" paradigm, periodically condensing exploration history into compact memory. Mem1 (Zhou et al., 2025b) synergizes memory and reasoning for efficient long-horizon agents, and Reasoning Path Compression (Song et al., 2025) compresses intra-trajectory generations to reduce inference cost. Reflexion (Shinn et al., 2023) adds verbal self-reflection between attempts when a reliable external evaluator is available. A critical distinction is that all of the above are *intra-trajectory* mechanisms (compressing or rewriting the context of a single rollout) or rely on an external evaluator. **Re-TRAC is fundamentally cross-trajectory**: each rollout finishes, a structured state (evidence, uncertainties, failures, future plans) is emitted as a coherent unit, and the next rollout starts from that state. The motivating problem is also different – not context length per se, but *incomplete branch exploration* across rollouts (Table 1). Empirically, on o3 in a training-free setting, Re-TRAC outperforms ReSum and IterResearch on BrowseComp300 (51.9 / 62.7 vs. 69.8), BrowseComp-ZH (59.3 / 60.5 vs. 64.6), and GAIA (70.5 / 73.8 vs. 75.0). Re-TRAC also avoids the dependence on an external summarizer that ReSum requires, and unlike IterResearch it operates as a meta-layer on top of any ReAct agent rather than replacing the inner loop.

## 2.3. Test-Time Scaling

While traditional scaling laws have focused on increasing model parameters and training data, recent paradigms have shifted towards test-time compute scaling (Wei et al., 2022; OpenAI, 2025b; Liu et al., 2025a; Du et al., 2023; Wang et al., 2022). The dominant approach to test-time scaling involves expanding the model's internal reasoning process. Chain-of-Thought (CoT) extensions (Wei et al., 2022) and reasoning models like OpenAI-o3 (OpenAI, 2025b) and DeepSeek-R1 (Guo et al., 2025) incentivize extended internal traces to decompose problems. A parallel direction

scales compute via ensemble strategies and inter-agent verification. Self-Consistency effectively marginalizes out reasoning errors by sampling diverse reasoning paths and applying majority voting to select the most robust answer (Wang et al., 2022). Multi-Agent Debate enables separate LLM instances to critique and refine each other's responses, leveraging adversarial dynamics to improve factuality and reduce hallucinations (Du et al., 2023). Recent work also studies parallel inter-dependent generations – Multiverse (Yang et al., 2025) lets language models decide how to parallelize and merge generations, Parallel-R1 (Zheng et al., 2025) learns parallel thinking via reinforcement learning, Generalized Parallel Scaling (Dong et al., 2025) extends parallel scaling with interdependent generations, and ParallelMuse (Li et al., 2025b) introduces agentic parallel thinking for deep information seeking – but all maintain a single round of parallel exploration without sequential information sharing across rounds. Our work introduces a sequential dimension to test-time scaling that differs from the parallel nature of voting or debate. We devise a novel mechanism to catalyze continuous self-reflection, enabling the model to explore a broader spectrum of possibilities with high computational efficiency. As shown in our experiments, Re-TRAC is moreover complementary to parallel TTS: a small number of Re-TRAC rounds combined with parallel aggregation outperforms pure parallel TTS at lower total compute.

## 3. Motivation

Through a systematic analysis of LLMs in Deep Research tasks, we identify two fundamental limitations that hinder performance. First, current models suffer from insufficient exploration, often converging prematurely on sub-optimal reasoning paths. While a naive solution to encourage exploration is to allow multiple trials (e.g., majority voting or Best-of-N), this introduces a secondary challenge: informational efficiency. The core problem lies in how to efficiently leverage these diverse trajectories to synthesize a superior final output.

**Incomplete Branch Exploration.** In order to find the bottleneck of current advanced deep research agents, we collect and analyze their trajectories where they fail to output correct answers. The analysis reveals a **common phenomenon**: in most failed trajectories, there are branches that the model plans to explore but forgets to explore in the end. As shown in Table 1, the ratio of this case can be up to 93%. We attribute this pervasive under-exploration to the *fundamental structural mismatch between the long-horizon nature of deep research tasks and the inherent linearity of the Re-Act framework*. While deep research necessitates strategic branching and backtracking, the ReAct paradigm constrains the agent to a sequential execution path, creating a discrepancy that inhibits the model's ability to pivot or re-evaluate

earlier decisions. Deep research tasks typically demand extended trajectories, which often span hundreds of thousands of tokens, characterized by a high density of interdependent tool calls. We observe that within the constraints of the linear ReAct framework, LLMs exhibit catastrophic forgetting as the trajectory lengthens. This is primarily because the model struggles to maintain long-term planning coherence. The critical task-level objectives formulated in the early stages are often marginalized by the accumulating volume of intermediate tool calls and observations.

*Table 1.* Ratio of trajectories containing incomplete branches among all failed trajectories. Evaluated on BrowseComp (Wei et al., 2025). Details are in Appendix A.

| Model | GLM-4.7 | DeepSeek-V3.2 | Tongyi-DeepResearch |
|-------|---------|---------------|---------------------|
| Ratio | 93.0%   | 92.7%         | 83.4%               |

**The Potential from Multiple Trials.** A straightforward approach to encourage exploration is through multiple stochastic trials (Wei et al., 2025). To quantify the untapped potential of extensive exploration, we evaluate various LLMs using the Pass@K metric. As illustrated in Figure 2, the substantial gap between Pass@1 and Pass@8 performance reveals a significant performance ceiling that current models fail to reach in a single trajectory.

Our empirical observations suggest that many failures are not rooted in the inherent reasoning capabilities of LLMs, but rather in the absence of an effective exploration management mechanism. While existing paradigms such as Majority Voting and Best-of-N (Wei et al., 2025) allow for multiple trials, these attempts remain independent. This lack of inter-trajectory communication leads to two critical inefficiencies: first, it results in repeated and redundant exploration, which squanders computational resources; second, it precludes the possibility of cross-trajectory experience sharing, making it difficult for the model to synthesize a global optimum from isolated experiences. This motivates a trajectory-level recursive agent framework. Instead of starting each attempt from scratch, the model explicitly compresses previous trajectories into a comprehensive experience of verified information and a meticulous enumeration of incomplete branches. By incorporating this feedback into $K$ sequential executions, it may systematically solve the planning and context issues identified in our analysis.

## 4. Method: Re-TRAC Framework

We propose **Re-TRAC** (REcursive TRAjectory Compression), an iterative trajectory-level framework. It utilizes a standardized compression specification to summarize previous attempts and propagates this context across successive rollouts. This mechanism ensures that each rollout is both

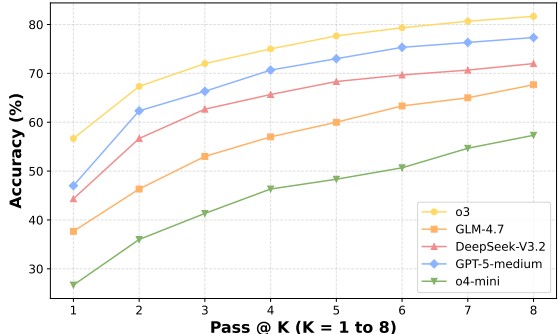

*Figure 2.* Performance of Pass@8 serves as the theoretical upper bound of the models.

efficient and informed by previous experiences. By continuously expanding the known search space, Re-TRAC effectively broadens plan coverage, reduces redundant exploration, and mitigates dead-end traps.

### 4.1. Trajectory Compression as a Structured State Representation

Figure 3 contrasts the standard ReAct paradigm (left) with our Re-TRAC framework (right). In ReAct, each rollout is a linear chain starting from the original query. Long contexts induce "**incomplete branch exploration**": as token count increases, early plans become less actionable, and the agent often loses track of decisive cues embedded in earlier observations. As illustrated in the left example, the agent may enumerate several candidate branches but fail to follow through, resulting in incomplete exploration coverage.

Re-TRAC solves these issues through *trajectory compression* (see Figure 4). After each rollout $t$, the trajectory $\tau_t$ is distilled into a structured state representation $S_t$. Following a fixed compression specification $\mathcal{C}$, the state is iteratively updated:

$$S_t \leftarrow Compress(\tau_t, S_{t-1}; \mathcal{C}). \qquad (1)$$

For deep research tasks, we define $S_t$ through three complementary facets that provide a comprehensive state representation for the agent:

- **Answer & Analytical Conclusions:** This facet records the best-supported partial answers. It also stores key inferences from the trajectory. Intermediate conclusions are kept as reusable anchors for later reasoning.

- **Evidence Base & Source Verification:** This facet records observed evidence and its provenance. It tracks which sources were consulted. It also marks which claims were verified. This helps avoid redundant tool calls and repeated checking.

- **Uncertainties & Exploration Trace:** This facet

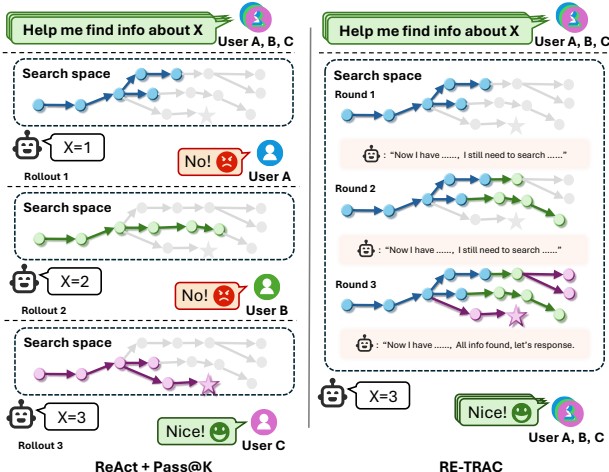

*Figure 3.* ReAct versus RE-TRAC framework. ReAct leads to premature convergence and forgotten branches in long-horizon tasks (left). RE-TRAC compresses experience from previous rollouts to systematically guide exploration in successive rounds (right).

records what remains unresolved. It includes open hypotheses and candidate branches. It also logs failed attempts and discarded directions. It helps the model to find unexplored search-space for the next rollout.

This structured state is added to the input of the subsequent rollout, ensuring that the agent starts each new attempt with a clear understanding of what is verified, what remains unresolved, and where to focus its exploration.

### 4.2. Recursive Execution with Structured State Representation

Re-TRAC is recursive by design. This process can be sustained over multiple rounds. The initial rollout functions like a standard ReAct execution. Consequently, it suffers from the same limitations, such as neglecting early planned branches.

The state representation acts as a guided search update. It filters out low-level trace details that unnecessarily consume context. Crucially, the state prevents exploration from collapsing into a single path. It explicitly retains multiple unresolved candidates and actionable options. This preserves branching diversity for subsequent rollouts. This balance between focused guidance and open branching supports controlled diversity throughout the recursion. Consequently, the agent progressively expands search space coverage while maintaining efficiency.

This recursive approach offers two primary benefits. First, it **improves coverage**. Incomplete branches are explicitly preserved and executed in later rollouts. Second, it **reduces redundancy**. The model avoids repeating tool calls for facts that are already verified. A branch missed in the first round

is captured in the state and directly explored in the next. In contrast, independent ReAct rounds often waste budget re-exploring the same path. Empirically, this compactness leads to higher efficiency in token usage and tool calls (see Section 5.3).

### 4.3. Application to Frontier Models

Re-TRAC functions as a training-free prompting strategy. It applies directly to frontier models during inference without fine-tuning. The execution process is straightforward:

First, we define a Deep Research Query and set a maximum round limit $N$ (defaulting to 8). In the initial round, the model utilizes the standard ReAct framework to generate a complete trajectory. We then compress this trajectory using a prompt (see in Appendix C.3.1) specifically designed to extract the Structured State Representation. The resulting state is used as the input for the next round. Specifically, it serves as the initial user message, positioned immediately after the system prompt. The model then performs another ReAct cycle to answer the question, building upon the previous state. This process repeats recursively until the round limit $N$ is reached. The answer generated in the final round serves as the final output of Re-TRAC.

In Section 5.3, we benchmark its performance against standard baselines, including Single Run, Best-of-$N$, Majority Voting, and Weighted Voting.

### 4.4. Training Small Models for Re-TRAC

Later experiments in Section 5.2 will show that deep research agents of different model sizes, ranging from 229B to 685B, can all benefit from the Re-TRAC workflow. This inspires us to explore whether a tiny edge model can achieve competitive performance, if equipped with the Re-TRAC workflow.

To investigate this, we distill Qwen3-4B-Instruct (Team, 2025b) and Tongyi-DeepResearch-30B-A3B (Team et al., 2025b) from GLM-4.7 on its Re-TRAC trajectories.

To obtain the raw prompts for training, we first construct a large amount of QA pairs via an entity-tree based method, following InfoAgent (Zhang et al., 2025). Specifically, we collect a large batch of entities from WikiPedia as roots of trees. Then for each entity, we recursively search its related entities as child nodes, until the tree grows up to pre-defined depth. The edges between neighbored nodes represents the relationship of the two entities. We synthesize a question by selecting a path from the root to a leaf node and converting the edges into sub-questions. In order to increase the difficulty of the questions, we also fuzzify the sub-questions using o3 (OpenAI, 2025b). Using this pipeline, we construct a total of 33K QA pairs. Then, we collect Re-TRAC (4 rounds) trajectories of GLM-4.7 on the

*Figure 4.* A comparative overview of the independent ReAct Pass@N (top) and our proposed RE-TRAC framework (bottom). Unlike the traditional ReAct paradigm, where multiple rollouts are executed in isolated silos without experience sharing, RE-TRAC is an iterative, trajectory-level framework. It employs a compression mechanism to distill analytical conclusions, evidence, and uncertainties from previous attempts. This compressed context is then propagated to successive rollouts, enabling the agent to recursively reflect on its trajectory and progressively improve its exploration strategy.

synthesized questions, resulting in 104k training samples after filtering, which are used to train our RE-TRAC-4B and RE-TRAC-30B-A3B models via SFT. Details are in Appendix B.

## 5. Experiments

### 5.1. Main Results

As shown in Table 2, we evaluate our RE-TRAC models on five challenging search-oriented benchmarks: BrowseComp (Wei et al., 2025), BrowseComp-ZH (Zhou et al., 2025a), XBench (Chen et al., 2025b), GAIA (Mialon et al., 2023), and HLE (Phan et al., 2025). A diverse set of competitive models are selected as baselines, which are grouped into four main categories: (1) Closed-Source Models, (2) Large Open-Source Models with more than 70B parameters, (3) Intermediate-sized Open-Source Models with 15B∼70B parameters, (4) Compact Open-Source Models with less than 15B parameters. Evaluation details are illustrated in Appendix C.

**Note on inference frameworks.** Several recent baselines (Chen et al., 2025a; Li et al., 2025a; OpenBMB, 2025) report results under specialized multi-round inference frameworks rather than a single Pass@1 ReAct rollout. We mark all such systems (including Re-TRAC) with † in Table 2 so that "single-run model capability" is not conflated with "framework capability". For transparency we additionally report *RE-TRAC-4B (light)* – the minimum Re-TRAC configuration (one ReAct trajectory followed by one Re-TRAC round) – which already reaches 21.7% on BrowseComp at ∼1.7× Pass@1 cost, surpassing InfoAgent-14B and WebExplorer-8B; the full inference-budget breakdown is given in Appendix D.1.

**Dominance in the Open-Source Models with the same size.** The primary finding from our evaluation is that our RE-TRAC models establish a new state-of-the-art among baselines of the same model size. Our RE-TRAC-30B-A3B model consistently achieves 7%∼11% improvement on BrowseComp, BrowseComp-ZH, GAIA and XBench, compared with its base model Tongyi-DeepResearch, which is also the previous strongest baseline with 30B parameters. For RE-TRAC-4B, it presents the best performance among all the benchmarks, compared with all the baselines with less than 15B parameters. These results demonstrate the superior advantages of our Re-TRAC framework.

**Competitive Performance Against Larger Models.** Beyond achieving the best performance among baselines of the same size, Re-TRAC shows competitive ability against much larger models. Notably, our 30B model beats MiniMax-M2-229B on all the benchmarks except HLE. On BrowseComp, its accuracy (53%) also exceeds that of GLM-4.7-358B (52%). This result indicates that the RE-TRAC framework is able to compensate for the lack of model intelligence by manually expanding its search space.

**Exceed Closed-Source Models.** While recent open-source large models (Kimi-K2, DeepSeek-V3.2) show a tendency to exceed the advanced closed-source models (GPT-5, Gemini-3-pro), our RE-TRAC-30B-A3B maintains this advantage, representing small open-source models. Specifically, it beats all the closed-source baselines on GAIA, and is also among the top results on BrowseComp and BrowseComp-ZH. This promising result implies that small models equipped with the Re-TRAC framework can replace those expensive proprietary products, serving as advanced on-device search agents.

### 5.2. Re-TRAC as a Test-Time Scaling Method

Previous experiments prove that the capacity of small models can be extended by training with Re-TRAC trajectories. In this section, we demonstrate that Re-TRAC can serve as a powerful and efficient training-free test-time scaling method for most general models on deep research tasks.

We implement the Re-TRAC framework for o4-mini (OpenAI, 2025b), o3 (OpenAI, 2025b), GPT-5 (Singh et al., 2025), DeepSeek-V3.2 (Liu et al., 2025a), and GLM-4.7 (zai org, 2025). Test-time scaling (TTS) methods for comparison follow the BrowseComp (Wei et al., 2025) setting:

- Re-TRAC: Select the final answer of the model after running $n$ RE-TRAC rounds (default $n$=8).

- Majority Voting (MV@n): Select the most frequent answer among $n$ independent solutions.

- Weighted Voting (WV@n): For $n$ independent solutions, we prompt the model to give a confidence score for its answer and weight each vote by the model's confidence in that answer. The answer with the most weighted votes is selected.

- Best-Of-N (Best@n): For $n$ independent solutions, we prompt the model to give a confidence score for its answer. The answer with the highest score is selected.

*Table 2.* Evaluation on deep research benchmarks. Here we evaluate on the BrowseComp full set. Accuracy(%) is reported according to existing studies. **Bold** indicates the best performance among models with the same size. Models marked with † use specialized inference frameworks that go beyond a single Pass@1 ReAct run (e.g., Markovian reconstruction in IterResearch, nested sub-agent calls in NestBrowse, multi-round recursive state transfer in Re-TRAC, multi-round browse iteration in AgentCPM-Explore); raw cross-row comparisons therefore conflate framework capability with single-run model capability. "Re-TRAC-4B (light)†" is the minimum Re-TRAC configuration (one ReAct trajectory followed by one Re-TRAC round) and is included to illustrate Re-TRAC's behavior at near-Pass@1 inference cost.

| Model | BrowseComp | BrowseComp-zh | GAIA | XBench | HLE |
|---|---|---|---|---|---|
| **Closed-Source Models** | | | | | |
| Claude-4.5-Sonnet (Anthropic, 2025) | 24.1 | 42.4 | 71.2 | 66.0 | 32 |
| o3 (OpenAI, 2025b) | 49.7 | 58.1 | 70.5 | 66.7 | 24.9 |
| OpenAI DeepResearch (OpenAI, 2025a) | 51.5 | 42.9 | 67.4 | - | 26.6 |
| GPT-5-high (Singh et al., 2025) | 54.9 | 63.0 | 76.7 | 77.9 | 42 |
| Gemini-3-pro (DeepMind, 2025) | 37.8 | 51.6 | 74.8 | - | 38.3 |
| **Large Open-Source Models (> 70B)** | | | | | |
| Kimi-K2-Thinking-1T (moonshotai, 2025) | 60.2 | 62.3 | - | - | 51.0 |
| DeepSeek-V3.2-Thinking-685B (Liu et al., 2025a) | 67.6 | 65.0 | - | - | 40.8 |
| GLM-4.7-358B (zai org, 2025) | 52.0 | 66.6 | - | - | 42.8 |
| MiniMax-M2-229B (MiniMax, 2025) | 44.0 | 48.5 | 75.7 | 72.0 | 31.8 |
| **Intermediate-sized Open-Source Models (15B∼70B)** | | | | | |
| Tongyi-DeepResearch-30B-A3B (Team et al., 2025b) | 43.4 | 46.7 | 70.9 | 75.0 | **32.9** |
| IterResearch-30B-A3B† (Chen et al., 2025a) | 37.3 | 45.2 | 72.8 | - | 28.8 |
| WebSailor-V2-30B-A3B (RL) (Li et al., 2025c) | 35.3 | 44.1 | 74.1 | 73.7 | 30.6 |
| **RE-TRAC-30B-A3B (Ours)†** | **53.0** | **57.3** | **78.2** | **83.0** | 31.5 |
| **Compact Open-Source Models (< 15B)** | | | | | |
| InfoAgent-14B (Zhang et al., 2025) | 15.3 | 29.2 | - | 40.4 | - |
| WebExplorer-8B (Liu et al., 2025b) | 15.7 | 32.0 | 50.0 | 53.7 | 17.3 |
| AgentCPM-Explore-4B† (OpenBMB, 2025) | 25.0 | 29.0 | 63.9 | 70.0 | 19.1 |
| NestBrowse-4B† (Li et al., 2025a) | 22.4 | 28.4 | 68.9 | 74.0 | - |
| **RE-TRAC-4B (light, Ours)†** | 21.7 | 26.8 | 60.9 | 64.9 | - |
| **RE-TRAC-4B (Ours)†** | **30.0** | **36.1** | **70.4** | **76.6** | **23.5** |

*Table 3.* Performance of different test-time scaling methods on BrowseComp300. Pass@1 is the basic method that the models only have one chance to solve the problem. To ensure fair comparison, all models are evaluated with the common self-hosted search and browse tools, thus the Pass@1 scores can be different with the officially reported scores.

| Model | Pass@1 | Re-TRAC | MV@8 | WV@8 | Best@8 |
|---|---|---|---|---|---|
| o4-mini | 25.7 | **46.8** | 34.0 | 46.7 | 43.3 |
| o3 | 54.9 | **69.8** | 64.3 | 70.0 | 68.0 |
| GPT-5-medium | 48.3 | **66.6** | 61.7 | 64.7 | 54.0 |
| DeepSeek-V3.2 | 45.3 | **60.8** | 55.7 | 57 | 55 |
| GLM-4.7 | 37.7 | **60.7** | 41.7 | 48 | 42.3 |

In order to save inference cost, for ablation experiments, we randomly sample 300 questions from BrowseComp as the test cases, constituting BrowseComp300. We empirically find that the model performance on this subset is very close to its performance on the full set.

**SOTA Performance.** As shown in Table 3, Re-TRAC achieves the best or competitive performance among all the models. Notably, advanced models (o4-mini, o3, GPT-5, DeepSeek-V3.2) can obtain significant gains from all these scaling methods, while GLM-4.7 fails to get comparable improvement via Majority Voting and Best-Of-N. This implies a gap between GLM-4.7 and the other models, in terms of the ability of self-judgment. Under this condition, the fact that all models can benefit from Re-TRAC demonstrates that Re-TRAC is a more general TTS method, and has loose requirements for the intelligence of models. The same trend extends across BrowseComp-ZH, DeepSearchQA (Gupta et al., 2026), and HLE, where Re-TRAC matches or exceeds all 8×-compute parallel baselines (including the concurrent ParallelMuse (Li et al., 2025b)) at ∼4× compute; full per-benchmark numbers and a Re-TRAC+parallel hybrid analysis are deferred to Appendix D.2.

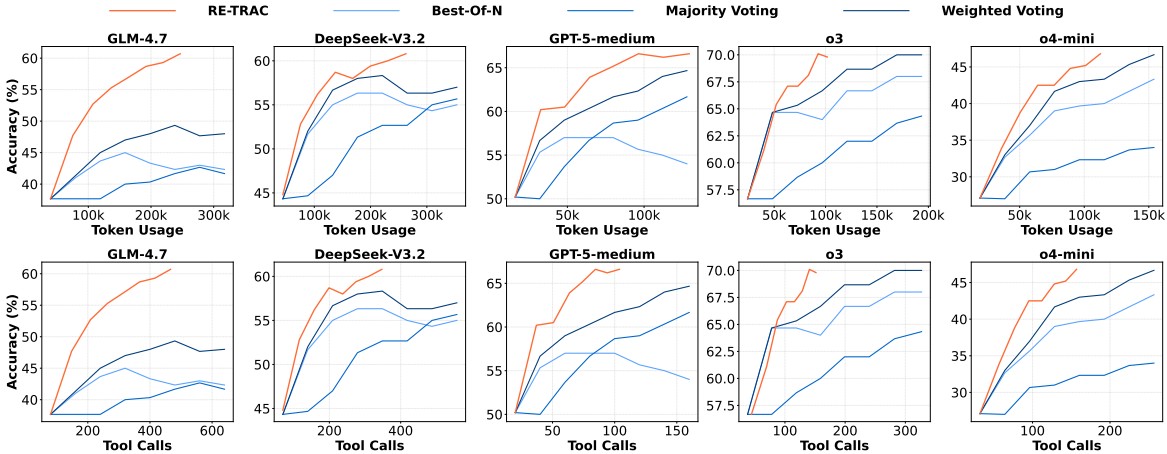

*Figure 5.* Relationship between model accuracy and used tokens/tools with different TTS methods. Evaluated on BrowseComp300. Re-TRAC consistently achieves better performance with less resources used.

**Economical Spending.** Figure 5 exhibits the token usage and tool usage of o3 under different TTS framework. For traditional TTS frameworks, since trajectories are independent of each other, the resource usage increases linearly when scaling up. For Re-TRAC, the model inherits states from previous rounds and its search space converges. Thus, redundant tool calls and exploration can be largely reduced when scaling up. Depending on task search-intensity, this enables Re-TRAC to save 25–50% of the resources required by 8×-compute parallel TTS at comparable or better accuracy (see Table 12).

**Comparison with context-management baselines.** Re-TRAC is conceptually adjacent to two recent intra-trajectory context-management approaches, IterResearch (Chen et al., 2025a) and ReSum (Wu et al., 2025). To make the distinction concrete, we evaluate all three frameworks on the same o3 backbone in a fully training-free setting. Re-TRAC reaches 69.8 / 64.6 / 75.0 on BrowseComp300 / BrowseComp-ZH / GAIA, versus 62.7 / 60.5 / 73.8 for Iter-Research and 51.9 / 59.3 / 70.5 for ReSum. Both baselines compress *within* a single rollout: IterResearch reconstructs task status step-by-step under a Markov assumption, which fragments cross-step dependencies, and ReSum periodically truncates and summarizes when the context window fills. Re-TRAC instead emits a coherent structured state *between* rollouts so each new trajectory begins from an explicit account of evidence, uncertainties, failures, and remaining plans. Unlike ReSum, Re-TRAC requires no external summarizer; unlike IterResearch, it does not replace the inner ReAct loop and therefore stacks cleanly on top of any existing agent.

## 5.3. Ablations

**Effect of SFT Training.** In previous experiments, we use SFT to train Qwen3-4B-Instruct and Tongyi-DeepResearch on the Re-TRAC trajectories generated by GLM-4.7, producing our RE-TRAC series models. A natural question is whether the resulting gains come primarily from *knowledge distillation* from GLM-4.7, or from learning the structured Re-TRAC trajectory format itself. To disentangle these factors, we compare four 4B variants in Table 4: (a) the base Qwen3-4B-Instruct under ReAct or under Re-TRAC at inference; (b) *Standard SFT* – distilled on GLM-4.7's standard ReAct trajectories (no Re-TRAC format), evaluated under both ReAct and Re-TRAC at inference; (c) RE-TRAC-4B (light), the same RE-TRAC-4B checkpoint but evaluated at the minimal Re-TRAC configuration (one Re-Act + one Re-TRAC round) for a ∼1.7× Pass@1 inference cost; (d) RE-TRAC-4B (default). Same teacher, same 33K questions, same hyper-parameters (Table 7) – only the trajectory format differs between Standard SFT and RE-TRAC SFT.

Two observations emerge. First, knowledge distillation alone (Standard SFT) brings Qwen3-4B from $1.0\%$ to $15.7\%$ on BrowseComp, confirming that GLM-4.7's knowledge is a meaningful contributor. Second, however, applying Re-TRAC at inference on top of Standard SFT yields only a marginal gain ($15.7 \rightarrow 16.2\%$), whereas training on Re-TRAC trajectories nearly doubles the score ($15.7 \rightarrow 30.0\%$). In other words, small models cannot leverage the Re-TRAC framework simply by being prompted with it: *the structured state generation and consumption behavior must be explicitly learned during training.* Even at the light configuration (one ReAct + one Re-TRAC round), RE-TRAC-4B (light) already reaches $21.7\%$, demonstrating that the gain is not merely an artifact of additional inference rounds.

*Table 4.* Disentangling knowledge distillation, Re-TRAC trajectory format, and number of Re-TRAC rounds. All 4B variants share the same teacher (GLM-4.7), the same 33K-question training set, and identical SFT hyper-parameters (Table 7); only the trajectory format and the inference-time framework differ. BC and BC-ZH are BrowseComp(300) and BrowseComp-ZH. "Re-TRAC" uses the default $K{=}8$ rounds; "light" denotes the minimum configuration with one ReAct trajectory followed by one Re-TRAC round.

| Model (inference framework) | BC | BC-ZH | GAIA | XBench | HLE |
|---|---|---|---|---|---|
| Qwen3-4B-Instruct (ReAct) | 1.0 | 3.1 | 17.3 | 48.0 | – |
| Qwen3-4B-Instruct (Re-TRAC) | 2.7 | 6.9 | 24.4 | 45.0 | 7.0 |
| Standard SFT (ReAct) | 15.7 | 17.9 | 46.2 | 56.0 | – |
| Standard SFT (Re-TRAC) | 16.2 | 17.6 | 51.3 | 55.0 | – |
| RE-TRAC-4B (light) | 21.7 | 26.8 | 60.9 | 64.9 | – |
| RE-TRAC-4B | **30.0** | **36.1** | **70.4** | **76.6** | **23.5** |

*Table 5.* Performance of o3 on BrowseComp300 with and without the prompt instructing the model to freely use the summary. Numbers in the brackets mean accuracy gains compared with previous round.

| Round | w/o free-use | w/ free-use |
|---|---|---|
| 1 | 56.1 | 56.1 |
| 2 | 61.2 (+5.1) | 63 (+7.0) |
| 3 | 64.0 (+2.8) | 65.7 (+2.7) |
| 4 | 66.4 (+2.4) | 67.0 (+1.3) |
| 5 | 66.8 (+0.4) | 69.3 (+2.3) |
| 6 | 68.2 (+1.4) | 70.0 (+0.7) |
| 7 | 68.5 (+0.3) | 71.0 (+1.0) |
| 8 | 68.9 (+0.4) | 71.7 (+0.7) |

This separates three distinct contributions: knowledge distillation ($1.0 \rightarrow 15.7\%$), the Re-TRAC trajectory format ($15.7 \rightarrow 21.7\%$), and additional Re-TRAC rounds at inference ($21.7 \rightarrow 30.0\%$). Table 4 also reveals that Re-TRAC SFT recovers a strong search agent through simple SFT, achieving comparable or even better performance than baselines trained by large-scale reinforcement learning (InfoAgent, WebExplorer in Table 2).

**Instructions for Utilizing Summary** In our early experiments, we find that the model can over-rely on the summary of previous rounds and get stuck in previous search path, failing to explore other branches. Hence, we explicitly instruct the model to judge whether the summary is valuable, and expand the search space as much as possible, allowing it to use the summary freely.

Table 5 shows that with the free-use prompt, the model outperformed in each round compared with the one without the prompt. This result is consistent with our analysis in Section 3 which proposes that the model can get stuck in some search path, failing to explore other possible branches if no other guidance is given.

**Quality of state representations** In each round of RE-TRAC, the model adjusts its search space according to the

*Table 6.* Effect of Summarizer on the final performance. Evaluated on BrowseComp300 with 8 rounds. Self means use the model itself as summarizer, which is default setting of RE-TRAC.

| Model/Summarizer | Self | GLM-4.7 |
|---|---|---|
| RE-TRAC-4B | 30.0 | 38.5 |
| RE-TRAC-30B-A3B | 53.0 | 52.4 |

state representations of previous rounds. Hence, the quality of the state representations can affect the final success rate.

To investigate this, we evaluate the performance of RE-TRAC models when using GLM-4.7 as the summarizer, as shown in Table 6. The results demonstrate that the 4B model can achieve better accuracy when using a stronger summarizer, while the 30B model has no improvement. This indicates that the summarization ability of our 4B model is relatively weak, and its search abilities have not been fully stimulated. We leave training a search agent with stronger summarization ability to future work.

## 6. Limitations and Future Work

Re-TRAC trades wall-clock latency for compute efficiency: under abundant parallelism, parallel TTS can be faster end-to-end despite higher total compute, while Re-TRAC matches accuracy at $\sim 4\times$ compute but $\sim 4\times$ longer single-query latency. Under resource-constrained deployment this trade-off reverses (Appendix D.3). The framework also depends on summarization quality, which is why small models require Re-TRAC-aware SFT (Table 4). Our SFT recipe currently relies on a strong teacher (GLM-4.7); self-training – a Re-TRAC-aware model generating and filtering its own trajectories – is a promising direction.

## 7. Conclusion

We presented Re-TRAC, a recursive, experience-based exploration framework that moves beyond linear ReAct for deep research agents. Cross-trajectory knowledge consolidation lets agents navigate complex search spaces with higher precision and lower compute. Consistent gains on frontier models, together with our SFT recipe for small models, highlight the importance of structured memory and conditioned planning.

## Impact Statement

This paper presents work whose goal is to advance the field of Machine Learning. There are many potential societal consequences of our work, none which we feel must be specifically highlighted here.

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

# A. Analysis Details of Incomplete Branch Exploration

In Section 3, we find that most deep research agents fail to explore all the proposed branches. This analysis is conducted by collecting their trajectories with incorrect final answers and prompting GPT-5 (Singh et al., 2025) to classify the trajectories. The used prompt is shown in Figure 6.

# B. Training Details

In Section 4.4, we distill Qwen3-4B-Instruct (Team, 2025b) and Tongyi-DeepResearch-30B-A3B (Team et al., 2025b) from GLM-4.7. We use GLM-4.7 to solve 33k questions from the dataset of InfoAgent (Zhang et al., 2025), via our Re-TRAC framework with 4 rounds. Since the context of each round is independent of the others, the solution trajectory for each question can be flattened to 4 training samples. Hence, we obtain 132k raw samples. We drop the samples that satisfy any of the following conditions: 1) The sample contains invalid tool calls. 2) The turn count is less than 15. 3) The sample does not have a valid final answer. This filtering strategy finally results in 104k high-quality training samples. Table 7 lists detailed hyper-parameters.

*Table 7.* Hyper-parameters for Supervised Fine-Tuning

| Setting | Value |
| --- | --- |
| number of samples | 104k |
| learning rate | 2e-5 |
| batch size | 512 |
| max length | 65536 |
| warmup ratio | 0.05 |
| learning rate scheduler | constant |
| weight decay | 0.1 |
| Adam $\beta_1$ | 0.9 |
| Adam $\beta_2$ | 0.95 |

# C. Evaluation Details

## C.1. Tools

We implement two primary tools for web-based retrieval and information extraction: search and visit. To ensure fair comparison, we adapt the tool interface to match each model's native function calling conventions, including parameter names, input types, and output formats.

### C.1.1. SEARCH TOOL

The search tool accepts a query string, performs web searches using Google Search Web API, and returns 5 relevant results per query. Each result includes the page title, URL, and a snippet of text. We didn't implement any additional processing or filtering on the search results.

For Tongyi models, the query parameter accepts a list of strings for batch search queries.

### C.1.2. VISIT TOOL

The visit tool accepts a list of URLs (parameter "urls", list of strings) and a goal string (parameter "goal", string), then fetches web pages, extracts their main textual content, and summarizes information relevant to a specified goal. For HTML pages, we use Trafilatura (Barbaresi, 2021) for text extraction. For PDF files, we extract text content page by page. The extracted content is then processed by GPT-4o-mini to generate structured summaries with prompt in Figure 7.

For Tongyi models, the parameter is named "url" instead of "urls", and output includes explicit "Evidence in page" and "Summary" sections.

For GLM models, the tool is named "open" with parameters "url" (single string) and "pattern" instead of "goal". The output contains only the summary section in a condensed format.

## C.2. Verifier

We follow the evaluation procedure introduced in BrowseComp (Wei et al., 2025) to assess the correctness of model outputs. For each question, along with its ground truth and the model's final answer, we employ OpenAI o4-mini to determine the correctness of the final answer with respect to the ground truth, with the prompt in Figure 8.

## C.3. Re-TRAC Details

### C.3.1. STRUCTURED STATE REPRESENTATION

As described in Section 4, the core of Re-TRAC is the trajectory compression mechanism that generates a structured state representation after each round. We design two variants of the state representation prompt tailored to different model capabilities.

**Base Version (for Smaller and SFT Models)** The base state representation (Figure 9) follows a fixed compression specification $\mathcal{C}$ that captures five complementary facets:

- **Current Answer**: The best-supported partial answer identified so far, or "None" if no conclusive evidence exists.

- **Facts & Evidence Collected**: All factual items discovered during the trajectory, with source annotations indicating provenance and verification status.

- **Analysis & Conclusions**: Logical conclusions derived

from the evidence, explicitly linked to supporting facts.

- **Source Inventory & Verification Status**: An enumeration of all visited sources and their current verification status.

- **Uncertainties, Limitations, Gaps**: Unknown variables, data ambiguities, and failure modes blocking a final decision.

**Full Version with Audit Part (for Frontier LLMs)** For frontier LLMs with stronger instruction-following and summarization capabilities, we extend the base version with three additional audit facets (Figure 10):

- **Failed Attempts**: Specific plans or objectives that were abandoned, left unfinished, or resulted in no progress by the end of the rollout.

- **Uncompleted Proposals**: Potential leads (URLs, entities, data points, or keywords) surfaced in tool outputs but never pursued due to token limits, focus shifts, or accidental omission.

- **Discarded Possibilities**: Candidate answers or critical evidence that were discarded based on unverified assumptions, hallucinations, or logical leaps.

The state representation prompt is appended to the trajectory after the model reaches a stopping condition (either providing a final answer or hitting the context limit).

### C.3.2. Continuation Prompt

To enable cross-trajectory knowledge consolidation, we prepend the structured state representation from the previous round to the input of the subsequent round, along with a continuation prompt that guides the model's utilization of this information. The continuation prompt template is shown in Figure 11.

### C.3.3. Round Settings

In our experiments, we evaluate Re-TRAC with varying numbers of rounds. Unless otherwise specified, we set the maximum round limit $K = 8$ as the default setting for test-time scaling experiments. For SFT data construction, we collect trajectories with $K = 4$ from GLM-4.7.

### C.3.4. Model-Specific Hyper-parameters

We apply Re-TRAC to multiple frontier and open-source models. Table 8 summarizes the key hyper-parameters used for each model during inference.

*Table 8.* Model-specific inference hyper-parameters for evaluation.

| Model | Context Window | Temperature | Top P | Reasoning |
|---|---|---|---|---|
| o4-mini | 200k | - | - | medium |
| o3 | 200k | - | - | medium |
| GPT-5 | 400k | - | - | medium |
| DeepSeek-V3.2 | 140k | 1.0 | 0.95 | enabled |
| GLM-4.7 | 128k | 1.0 | 0.95 | - |
| RE-TRAC-30B-A3B | 128k | 0.7 | 1.0 | - |
| RE-TRAC-4B | 128k | 0.7 | 0.8 | - |

## D. Additional Experiments

### D.1. Inference-Budget Breakdown for Table 2

Table 2 marks systems that use specialized multi-round inference frameworks with †. Table 9 summarizes the additional inference cost incurred by each system. By default Re-TRAC runs $K=8$ sequential rounds; because per-round tool usage decreases monotonically (Figure 5), the effective compute is roughly $4\times$ a single ReAct run rather than $8\times$. We additionally report RE-TRAC-4B (light), a 1+1 configuration that already reaches 21.7% on BrowseComp at $\sim 1.7\times$ Pass@1 cost.

*Table 9.* Approximate inference budgets for the specialized-framework rows in Table 2.

| System | Effective Inference Cost |
|---|---|
| IterResearch | Markovian reconstruction; varies with interaction limit |
| NestBrowse | ReAct + nested browse sub-agent (cost not disclosed) |
| AgentCPM-Explore | Multi-round browse loop |
| RE-TRAC (light, 2 rounds) | $\sim 1.7\times$ a single ReAct run |
| RE-TRAC (default, 8 rounds) | $\sim 4\times$ a single ReAct run |

### D.2. TTS Generalization and Hybridization

**Cross-benchmark TTS comparison.** Table 10 extends Table 3 from BrowseComp300 to three additional benchmarks, comparing Re-TRAC against parallel TTS baselines as well as the concurrent ParallelMuse (PM@8) (Li et al., 2025b) on o3. Re-TRAC matches or exceeds all $8\times$-compute parallel methods on 2 of 4 benchmarks at $\sim 4\times$ compute, and substantially outperforms them on DeepSearchQA (+6.9 over WV@8) and HLE (+2.1 over PM@8). The advantage is most pronounced on multi-entity tasks such as DeepSearchQA (Gupta et al., 2026), where the gold answer is an aggregated entity list: parallel runs independently discover different subsets but cannot merge findings, while Re-TRAC's cross-round structured state accumulates entities progressively.

*Table 10.* Test-time scaling on four benchmarks with o3.

| Benchmark | Pass@1 | MV@8 | WV@8 | Best@8 | PM@8 | Re-TRAC |
|---|---|---|---|---|---|---|
| BrowseComp300 | 54.9 | 64.3 | 70.0 | 68.0 | 70.3 | 69.8 |
| BrowseComp-ZH | 58.1 | 64.7 | 66.4 | 63.0 | 65.4 | 64.6 |
| DeepSearchQA | 58.2 | 60.7 | 63.7 | 61.7 | 63.7 | **70.6** |
| HLE | 24.9 | 29.3 | 31.0 | 31.7 | 35.0 | **37.1** |

**Re-TRAC composes with parallel TTS.** Because Re-TRAC's per-round cost decreases monotonically, a short Re-TRAC prefix combined with parallel aggregation can outperform pure parallel TTS at lower total compute (Table 11). BOP@8 denotes the maximum across MV@8, WV@8, Best@8, and PM@8; Re-TRAC(2)+BOP@4 runs two Re-TRAC rounds and applies a 4-way parallel aggregation at the second round.

*Table 11.* Hybrid sequential+parallel TTS on o3. BOP@8 = best of all parallel TTS at 8 trajectories; Re-TRAC(2)+BOP@4 uses strictly less total compute than BOP@8.

| Benchmark | BOP@8 | Re-TRAC(2)+BOP@4 |
|---|---|---|
| BrowseComp300 | 70.3 | **70.6** |
| BrowseComp-ZH | 66.4 | 66.4 |
| DeepSearchQA | 63.7 | **67.7** |
| HLE | 35.0 | **37.5** |

**Per-round cost decay.** Table 12 reports tool calls per round, accumulated accuracy, and accumulated wall-clock for o3 on BrowseComp300. Tool calls drop from 44.4 in round 1 to 10.8 in round 8 (a $-76\%$ reduction), so 8 sequential Re-TRAC rounds cost roughly $4\times$ rather than $8\times$ a single round.

*Table 12.* Re-TRAC per-round cost decay (o3, BrowseComp300, concurrency=10).

| Round | Acc% | Avg. Tool Calls | Wall-Clock (min) |
|---|---|---|---|
| 1 | 53.7 | 44.4 | 49 |
| 2 | 61.1 | 25.2 | 104 |
| 3 | 65.4 | 17.8 | 129 |
| 4 | 67.1 | 15.2 | 168 |
| 8 | 69.8 | 10.8 | 201 |

### D.3. Wall-Clock Latency Analysis

**Abundant-resource regime (API).** With the o3 API at concurrency=10 on BrowseComp300, parallel TTS launches 8 independent trajectories that overlap in wall-clock, while Re-TRAC must run rounds sequentially:

*Table 13.* Wall-clock comparison under abundant API parallelism.

| Method | Acc% | Compute | Wall-Clock |
|---|---|---|---|
| Pass@1 | 54.9 | $1\times$ | 49 min |
| MV@8 | 64.3 | $8\times$ | ~49 min |
| WV@8 | 70.0 | $8\times$ | ~49 min |
| Re-TRAC | 69.8 | $\sim4\times$ | 201 min |

**Resource-constrained regime (self-hosted GPUs).** Under a fixed GPU budget, parallel TTS launches $8\times$ as many

concurrent jobs and therefore queues for longer. We deploy Tongyi-DR-30B on $4\times$B200 with SGLang + HiCache (concurrency=100) over BrowseComp100:

*Table 14.* Wall-clock under finite GPU throughput.

| Completion | RE-TRAC (s) | Parallel TTS@8 (s) | Speedup |
|---|---|---|---|
| 75% | 2978 | 3969 | $1.33\times$ |
| 90% | 3601 | 4825 | $1.34\times$ |
| 100% | 4184 | 5973 | $1.43\times$ |

The two regimes capture the fundamental trade-off: parallel TTS wins on per-query latency when parallelism is unlimited, while Re-TRAC wins on throughput when parallelism is bounded. The two are also complementary (Table 11).

### D.4. Post-Training Branch Analysis

The motivation analysis in Table 1 found that 83.4% of Tongyi-DR-30B's failed BrowseComp trajectories contained planned-but-unexplored branches. Re-applying the same GPT-5-based classification (Figure 6) to RE-TRAC-30B-A3B's failed trajectories on BrowseComp drops this ratio to **52.5%**, confirming that Re-TRAC substantially alleviates the failure mode that motivated the framework. The residual 52.5% indicates that branch exploration is not yet fully addressed, suggesting that further gains may come from stronger summarization quality (cf. Table 6) or self-training on Re-TRAC trajectories.

### D.5. Prompt Sensitivity

A reasonable concern is whether Re-TRAC's gains stem from the framework or from its prompt. We apply Re-TRAC's prompt (denoted †) to o3 in standalone, MV@8, and WV@8 configurations on the same evaluation suite:

*Table 15.* Isolating the contribution of Re-TRAC's prompt (†) from the framework itself, on o3.

| Config | BC | BC-ZH | XBench | HLE |
|---|---|---|---|---|
| o3 | 49.7 | 58.1 | 66.7 | 24.9 |
| o3 MV@8 | 64.3 | 64.7 | – | 29.3 |
| o3 WV@8 | 70.0 | 66.4 | – | 31.0 |
| o3† | 55.7 | 56.8 | 79.0 | 28.7 |
| o3† MV@8 | 60.3 | 64.7 | 79.0 | 29.3 |
| o3† WV@8 | 68.0 | 66.4 | 82.0 | 31.0 |
| o3 Re-TRAC | **69.8** | **64.6** | **85.7** | **37.1** |

The prompt alone (row 1 vs. 4) adds $+6.0$ on BrowseComp; the full Re-TRAC framework (row 4 vs. 7) adds another $+14.1$. Interestingly, the same prompt slightly *hurts* parallel TTS (MV@8: $64.3 \rightarrow 60.3$; WV@8: $70.0 \rightarrow$

68.0). A fixed prompt – even a well-designed exploration-encouraging one – constrains generation behavior across the $N$ parallel trajectories, and prior work has shown that the resulting drop in sampling diversity erodes the gains of Best-of-N and majority voting (Wang et al., 2025). Re-TRAC's sequential cross-round state transfer does not depend on trajectory diversity in the same way, which is why the framework retains the gain.

## D.6. Standard SFT Configuration

The Standard SFT baseline in Table 4 uses identical training hyper-parameters to RE-TRAC SFT (Table 7): learning rate 2e-5, batch size 512, max length 65536, warmup ratio 0.05, constant LR scheduler, weight decay 0.1, Adam $\beta_1 = 0.9$, $\beta_2 = 0.95$. The training pool is the same 33K questions distilled from GLM-4.7; the only difference is that Standard SFT collects GLM-4.7's standard ReAct trajectories (no structured state). No hyper-parameter sweep was performed for either baseline, so the comparison isolates the trajectory format.

## D.7. Evaluation Results

The test-time scaling methods (RT@N, MV@N, WV@N, Best@N) are defined in Section 5.2. In addition to these metrics and Pass@N, we introduce one additional metric specific to evaluating Re-TRAC in this appendix:

**Accuracy Prefix** (AP@N): Similar to Pass@N, AP@N measures whether at least one correct answer appears among the first N rounds. AP@N serves as an upper bound for Re-TRAC performance and helps quantify how much room remains for improvement in answer selection strategies.

### D.7.1. PER-MODEL DETAILED RESULTS

We present detailed round-by-round results for each model evaluated on BrowseComp300. Each table reports the accuracy at each individual round (Acc%), cumulative Pass@N, Re-TRAC accuracy (RT@N), Accuracy Prefix (AP@N), Majority Voting (MV@N), Weighted Voting (WV@N), and Best-of-N (Best@N) across 8 rounds.

Results for o4-mini, o3, GPT-5, DeepSeek-V3.2, and GLM-4.7 are presented in Table 16, Table 17, Table 18, Table 19, and Table 20, respectively.

*Table 16.* Round-by-round evaluation results for o4-mini on BrowseComp300.

| Round / N | Acc% | Pass@N | RT@N | AP@N | MV@N | WV@N | Best@N |
|-----------|------|--------|------|------|------|------|--------|
| 1 | 26.7 | 26.7 | 26.7 | 26.7 | 26.7 | 26.7 | 26.7 |
| 2 | 26.1 | 36.0 | 33.8 | 34.1 | 26.7 | 32.7 | 32.7 |
| 3 | 25.3 | 41.3 | 38.8 | 39.1 | 29.7 | 36.3 | 36.0 |
| 4 | 24.0 | 46.3 | 42.5 | 42.8 | 29.0 | 40.7 | 39.3 |
| 5 | 25.5 | 48.3 | 42.5 | 43.1 | 29.7 | 42.0 | 40.0 |
| 6 | 25.1 | 50.7 | 44.8 | 45.5 | 29.0 | 43.0 | 40.3 |
| 7 | 26.3 | 54.7 | 45.2 | 46.5 | 29.0 | 42.3 | 42.0 |
| 8 | 26.4 | 57.3 | 46.8 | 47.8 | 30.3 | 44.7 | 43.7 |

*Table 17.* Round-by-round evaluation results for o3 on BrowseComp300.

| Round / N | Acc% | Pass@N | RT@N | AP@N | MV@N | WV@N | Best@N |
|-----------|------|--------|------|------|------|------|--------|
| 1 | 56.7 | 56.7 | 56.7 | 56.7 | 56.7 | 56.7 | 56.7 |
| 2 | 54.7 | 67.3 | 61.1 | 62.1 | 56.7 | 64.7 | 64.7 |
| 3 | 51.3 | 72.0 | 65.4 | 66.1 | 58.7 | 65.3 | 64.7 |
| 4 | 54.7 | 75.0 | 67.1 | 67.4 | 60.0 | 66.7 | 64.0 |
| 5 | 56.0 | 77.7 | 67.1 | 68.1 | 62.0 | 68.7 | 66.7 |
| 6 | 54.9 | 79.3 | 68.1 | 69.1 | 62.0 | 68.7 | 66.7 |
| 7 | 56.3 | 80.7 | 70.1 | 71.1 | 63.7 | 70.0 | 68.0 |
| 8 | 54.7 | 81.7 | 69.8 | 71.1 | 64.3 | 70.0 | 68.0 |

*Table 18.* Round-by-round evaluation results for GPT-5 on BrowseComp300.

| Round / N | Acc% | Pass@N | RT@N | AP@N | MV@N | WV@N | Best@N |
|-----------|------|--------|------|------|------|------|--------|
| 1 | 47.0 | 47.0 | 47.0 | 47.0 | 47.0 | 47.0 | 47.0 |
| 2 | 54.0 | 62.3 | 60.2 | 60.9 | 47.0 | 55.7 | 55.3 |
| 3 | 46.0 | 66.3 | 60.5 | 62.2 | 48.7 | 56.3 | 57.0 |
| 4 | 48.7 | 70.7 | 63.9 | 65.2 | 52.7 | 56.7 | 57.0 |
| 5 | 46.7 | 73.0 | 65.2 | 66.6 | 53.0 | 57.0 | 57.0 |
| 6 | 48.0 | 75.3 | 66.6 | 67.6 | 52.3 | 57.0 | 55.7 |
| 7 | 47.3 | 76.3 | 66.2 | 68.2 | 52.0 | 56.0 | 55.0 |
| 8 | 48.7 | 77.3 | 66.6 | 69.2 | 54.0 | 56.3 | 54.0 |

*Table 19.* Round-by-round evaluation results for DeepSeek-V3.2 on BrowseComp300.

| Round / N | Acc% | Pass@N | RT@N | AP@N | MV@N | WV@N | Best@N |
|-----------|------|--------|------|------|------|------|--------|
| 1 | 44.3 | 44.3 | 44.3 | 44.3 | 44.3 | 44.3 | 44.3 |
| 2 | 46.0 | 56.7 | 52.8 | 54.5 | 44.3 | 52.7 | 52.7 |
| 3 | 44.7 | 62.7 | 56.2 | 59.0 | 46.7 | 58.3 | 57.0 |
| 4 | 44.0 | 65.7 | 58.7 | 60.4 | 48.7 | 60.0 | 59.3 |
| 5 | 47.7 | 68.3 | 58.0 | 61.1 | 47.7 | 59.3 | 60.3 |
| 6 | 42.7 | 69.7 | 59.4 | 62.2 | 48.3 | 58.0 | 59.7 |
| 7 | 45.0 | 70.7 | 60.1 | 63.2 | 49.0 | 56.7 | 59.7 |
| 8 | 47.7 | 72.0 | 60.8 | 63.9 | 50.7 | 58.0 | 60.3 |

*Table 20.* Round-by-round evaluation results for GLM-4.7 on BrowseComp300.

| Round / N | Acc% | Pass@N | RT@N | AP@N | MV@N | WV@N | Best@N |
|-----------|------|--------|------|------|------|------|--------|
| 1 | 37.7 | 37.7 | 37.7 | 40.3 | 37.7 | 37.7 | 37.7 |
| 2 | 36.3 | 46.3 | 47.7 | 48.7 | 37.7 | 41.3 | 41.0 |
| 3 | 37.1 | 53.0 | 52.7 | 53.7 | 37.7 | 45.0 | 43.7 |
| 4 | 41.3 | 57.0 | 55.3 | 56.3 | 40.0 | 47.0 | 45.0 |
| 5 | 39.7 | 60.0 | 57.0 | 58.3 | 40.3 | 48.0 | 43.3 |
| 6 | 45.3 | 63.3 | 58.7 | 60.7 | 41.7 | 49.3 | 42.3 |
| 7 | 41.0 | 65.0 | 59.3 | 62.0 | 42.7 | 47.7 | 43.0 |
| 8 | 41.0 | 67.7 | 60.7 | 62.7 | 41.7 | 48.0 | 42.3 |

```
I will show you a solution trajectory of another model that try to find the answer to
a question via deep research.  The model fails to give an correct answer finally.  I
need you to analyze the trajectory and detect whether the model has one of the
following behaviors:

A. The model finally gives an answer or asks the user for more information.  It has
planned to try some other clues or possible branches during the process, but not all
proposed clues and branches are explored in the end.

B. The model finally gives an answer or asks the user for more information.  It has
explored all clues and branches proposed during its search process.

C. The model finally hits the context limitation.  It has planned to try some other
clues or possible branches during the process, but finally stucks in an unproductive
branch for a long time.

D. The model finally hits the context limitation.  It has planned to try some other
clues or possible branches during the process, and it is still exploring these
different branches until hitting the context limitation.

E. None of the above.

Note that when the context limitation is hit, the user will warn the model and force
the model to give an answer.

Your output should be in the following json format:

{
  "behavior":  <A, B, C, D or E>,
  "reason":  <A brief reason for your choice>
}

Here is the trajectory:

{traj}
```

*Figure 6.* Prompt for classifying trajectory failure modes in incomplete branch exploration analysis.

```
Please process the following webpage content and user goal to extract relevant
information:

## **Webpage Content**
{webpage_content}

## **User Goal**
{goal}

## **Task Guidelines**
1.  **Content Scanning for Rationale**:  Locate the specific sections/data directly
related to the user's goal within the webpage content
2.  **Key Extraction for Evidence**:  Identify and extract the most relevant
information from the content, output the full original context as far as possible
3.  **Summary Output for Summary**:  Organize into a concise paragraph with logical
flow, prioritizing clarity

**Final Output Format using JSON format has "rational", "evidence", "summary"
fields**
```

*Figure 7.* Prompt for the visit tool to extract and summarize goal-relevant information from webpage content.

```
You are an evaluator.  Based ONLY on the [correct_answer], judge whether the
[response] to the [question] is correct.

=== INPUTS ===
[question]:  {question}
[response]:  {answer}
[correct_answer]:  {ground_truth}

=== TASK ===
1.  Extract the single final answer from the [response].  If no clear final answer
exists, write "None".
2.  Give a concise explanation (reasoning) that ONLY compares the extracted answer
with the [correct_answer].  Do not solve the problem again or add extra background.
3.  Decide correctness:  set correctness = correct if they are equivalent / within a
tiny numeric tolerance and acceptable difference of expression style; otherwise
incorrect.  [correct_answer] may contain multiple answers separated by "OR", the
response is correct if it matches any of the answers.
4.  Extract a confidence score (0{100).  If the [response] provides none, use 100.

=== OUTPUT FORMAT (STRICT) ===
Return a valid JSON object with exactly these keys:
{
  "extracted_final_answer":  <string>,
  "reasoning":  <string>,
  "correctness":  <string "correct" or "incorrect">,
  "confidence":  <integer 0-100>
}

Do NOT output anything else|no comments, no code fences.
```

*Figure 8.* Prompt for answer verification using o4-mini, following the BrowseComp evaluation protocol.

```
--- TASK COMPLETED / STARTING SUMMARIZE ---
Summary the trajectory above for the original question you are given: {input}.

CRITICAL OUTPUT FORMAT REQUIREMENTS (STRICT)
You MUST follow the exact format below. Do NOT add extra sections, headers,
prefaces, or commentary.
Do NOT use placeholders like "see above", "as mentioned", "ibid.", "refer to
earlier", or "same as before".
Do NOT omit any important information that appears in the input.

OUTPUT FORMAT (EXACT)

You must directly record the actual information content in plain text (not just
summaries). Include:

0) Current Answer
  - Provide the single best, most definite answer identified so far based on
conclusive evidence. If no final answer is certain, state "None".

1) Facts & Evidence Collected
  - List every factual item discovered. Attach a source annotation: [Source:
<tool/doc> | <URL/id> | Verified: yes/no/partial].

2) Analysis & Conclusions
  - State all logical conclusions derived, explicitly linking each to the supporting
evidence IDs from section (1).

3) Source Inventory & Verification Status
  - Enumerate all visited sources and evaluate their current verification status.

4) Uncertainties, Limitations, Gaps
  - List all unknown variables, data ambiguities, and failure modes currently
blocking a final decision.

QUALITY RULES
- Be exhaustive over the input: no omissions.
- Keep claims tightly tied to evidence; if you infer, label it "Inference" and
 explain why.
- If sources conflict, present both and state which you trust more (and why).
```

*Figure 9.* Re-TRAC State Representation Prompt (base version without audit facets) used for smaller and SFT models.

```
--- TASK COMPLETED / STARTING SUMMARIZE ---
Summary the trajectory above for the original question you are given: {input}.

CRITICAL OUTPUT FORMAT REQUIREMENTS (STRICT)
You MUST follow the exact format below. Do NOT add extra sections, headers,
prefaces, or commentary.
Do NOT use placeholders like "see above", "as mentioned", "ibid.", "refer to
earlier", or "same as before".
Do NOT omit any important information that appears in the input.

OUTPUT FORMAT (EXACT)

You must directly record the actual information content in plain text (not just
summaries). Include:

0) Current Answer
  - Provide the single best, most definite answer identified so far based on
conclusive evidence. If no final answer is certain, state "None".

1) Facts & Evidence Collected
  - List every factual item discovered. Attach a source annotation: [Source:
<tool/doc> | <URL/id> | Verified: yes/no/partial].

2) Analysis & Conclusions
  - State all logical conclusions derived, explicitly linking each to the supporting
evidence IDs from section (1).

3) Source Inventory & Verification Status
  - Enumerate all visited sources and evaluate their current verification status.

4) Uncertainties, Limitations, Gaps
  - List all unknown variables, data ambiguities, and failure modes currently
blocking a final decision.

5) Failed attempts
  - Identify any specific plan or objective explicitly stated in the reasoning that
was either abandoned, left unfinished, or resulted in no progress by the end of the
rollout.

6) Uncompleted proposals
  - Scan all tool outputs and internal reasoning for potential leads (URLs, entities,
data points, or keywords) that were surfaced but never pursued due to token limits,
focus on a different branch, or accidental omission.

7) Discarded possibilities
  - Identify any candidate answers or critical evidence that were discarded based on
unverified assumptions, hallucinations, or logical leaps.

QUALITY RULES
- Be exhaustive over the input: no omissions.
- Keep claims tightly tied to evidence; if you infer, label it "Inference" and
explain why.
- If sources conflict, present both and state which you trust more (and why).
```

*Figure 10.* Re-TRAC State Representation Prompt (full version with audit facets) used for frontier LLMs.

```
Below is the summary from the previous attempt:

{last_summary}

1.  Critical Evaluation:
  The summary provided above is a *suggested* synthesis of past efforts, not an
absolute truth.  It may contain hallucinations, unverified assumptions, or premature
conclusions.  You are encouraged to:
  - Identify and ignore any parts of the summary that seem illogical or poorly
supported.
  - If the summary's quality is low or its direction feels like a dead-end, you have
the full autonomy to completely disregard it and initiate a fresh search strategy.

2.  Expand the search space:

  If the task remains unsolved, it means the current search space is insufficient.
Do not get trapped in the "logic loop" of the summary.  Use the [Uncertainties,
Limitations, Gaps] section as a springboard to:
  - Pivot to entirely different keyword clusters or tool-call strategies.
  - Cross-verify "Facts" that were marked as "unverified" or "partial" in the
summary.

3.  Autonomous re-planning:
  The summary provides the *memory*, but you provide the *reasoning*.  Based on the
existing facts and your own judgment of the current situation, determine your own
next steps.

Now continue your deep search task.  You have full permission to be skeptical of the
past and bold in your current exploration.
```

*Figure 11.* Re-TRAC Continuation Prompt prepended to subsequent rounds to guide cross-trajectory knowledge consolidation.

