# OpenReview forum: "RE-TRAC: REcursive TRAjectory Compression for Deep Search Agents"
_ICML.cc/2026/Conference — ICML 2026 regular_

### Official Review · Reviewer_DA8W · 2026-03-05

**Soundness:** 2
**Presentation:** 3
**Significance:** 2
**Originality:** 2
**Overall Recommendation:** 3
**Confidence:** 3

**Summary:**

The paper proposes Re-TRAC that mitigates the issues of deep research agents, including incomplete branch exploration and inefficient token-usage (majority voting). Re-TRAC divide a deep research task into several rounds and compresses the contexts in previous rounds. This recursive strategy improves 15-20% over ReAct on BrowseComp.

The authors further explore Re-TRAC-aware supervised fine-tuning for smaller models. Re-TRAC fine-tuned 4B and 30B models surpass the performance of significantly larger models (MiniMaxM2-229B).

**Compliance With Llm Reviewing Policy:**

Affirmed.

**Final Justification:**

The rebuttal does not address my concerns, namely, the computation overhead and the general competence of the proposed method. So I keep my score.

**Key Questions For Authors:**

1. Could you provide a comparison of the wall-clock time (latency) between Re-TRAC and Parallel Majority Voting?
2. For 4B/30B comparison with significantly larger models, could you justify the use of GLM-4.7 (weakness-3)?
3. Could you provide prompt ablations or at least justify your prompt setup for your method and the baselines?

**Limitations:**

Discussed in the weakness part

**Strengths And Weaknesses:**

**Strength**

1. The central idea of Re-TRAC is minimalist and very intuitive.
2. Re-TRAC achieves significant improvements on deep research benchmarks and even outperform large models with small models + fine-tuning.
3. Re-TRAC is scalable for long-context tasks with the context compression technique.
4. The experiments are systematic and sound.

**Weakness**

1. The main baselines include ReAct and majority voting (the authors mentioned parallel works like IterResearch and WebSailor as well). However, the authors did not justify the prompt difference of different approaches. For example, if we use strong and specific prompts for ReAct and explicitly forces it to explore forgotten branches, ReAct might have achieved better results. The authors need to justify the sensitivity of ReAct and their method upon different prompt setups.
2. There are other deep research relevant methods with long-term memory or similar iterative idea (e.g. Reflexion, IterResearch). But the core test-time scaling experiments did not include these related prior arts.
3. The authors use synthetic data from GLM-4.7 in SFT but necessary ablations are missing. They need to justify if the major performance improvements are caused by knowledge distillation from GLM-4.7. If the authors use "normal" synthetic data without Re-TRAC-aware tricks, the 4B/30B model might still reach similar performance. I am not sure if GLM-4.7 knowledge contributes more or Re-TRAC fine-tuning really matters.

---

> ### Author Rebuttal · Authors · 2026-03-31
>
> We thank the reviewer for:
> - recognizing the minimalist and intuitive design of Re-TRAC
> - acknowledging the significant improvements and small-model outperforming large-model results
> - appreciating the scalability for long-context tasks
>
> The reviewer raises concerns on:
> - W1 & Q3: prompt sensitivity
> - W2: missing comparisons with Reflexion/IterResearch
> - W3 & Q2: GLM-4.7 distillation ablation
> - Q1: wall-clock latency
>
> ### DA8W-W1 & Q3: Prompt Sensitivity / Ablation
>
> > *"If we use strong prompts for ReAct to explore forgotten branches, ReAct might have achieved better results."*
>
> **The prompt contributes only +6.0% on BC, while the full Re-TRAC framework contributes +14.1% — the performance gains are driven by cross-trajectory structured state, not prompt engineering.**
>
> Re-TRAC does use a different prompt from standard ReAct (encouraging broader exploration). To fairly isolate this factor, we apply Re-TRAC's prompt (denoted †) to o3 — a frontier model with strong reasoning and instruction-following capabilities — in both single-pass and parallel TTS:
>
> |Config|BC|BC-ZH|XBench|HLE|
> |---|---|---|---|---|
> |o3|49.7|58.1|66.7|24.9|
> |o3 MV@8|64.3|64.7|—|29.3|
> |o3 WV@8|70.0|66.4|—|31.0|
> |o3†|55.7|56.8|79.0|28.7|
> |o3† MV@8|60.3|64.7|79.0|29.3|
> |o3† WV@8|68.0|66.4|82.0|31.0|
> |**o3 Re-TRAC**|**69.8**|**64.6**|**85.7**|**37.1**|
>
> The † prompt improves single-pass by +6.0% on BC (row 1 vs 4), but the full Re-TRAC framework adds +14.1% (row 4 vs 7). Notably, † *hurts* parallel TTS: MV@8 drops 64.3 to 60.3 (row 2 vs 5), WV@8 drops 70.0 to 68.0 (row 3 vs 6). The structured prompt constrains generation behavior, reducing trajectory diversity that parallel TTS relies on — consistent with recent findings on diversity collapse under fixed prompts (Wang et al., 2025, arXiv:2502.11027). Re-TRAC's sequential architecture does not depend on trajectory diversity; cross-round state transfer serves a fundamentally different role. Table 5 further confirms Re-TRAC is robust to prompt variation (+2.8%). The gap is structural, not prompt-driven.
>
>
> ### DA8W-W2: Missing Reflexion/IterResearch
>
> > *"There are other methods with long-term memory or iterative ideas."*
>
> We agree these are relevant related works. Table 3 does not include them because **they are not trajectory-level TTS methods with controllable round budgets** — Table 3 methods (MV@8, WV@8, Best@8, Re-TRAC@8) all run exactly 8 rounds. IterResearch uses a Markovian reconstruction process where the number of interactions is not fixed by round count, and Reflexion requires a reliable evaluator signal that is difficult to obtain in deep research tasks.
>
> Nonetheless, we provide a direct comparison on o3 across three benchmarks (see LSZ8-W1): Re-TRAC (69.8/64.6/75.0) consistently outperforms IterResearch (62.7/60.5/73.8) and ReSum (51.9/59.3/70.5) on BC300/BC-ZH/GAIA. We will expand Related Work to discuss these connections.
>
> ### DA8W-W3 & Q2: SFT Ablation
>
> > *"They need to justify if improvements are caused by knowledge distillation from GLM-4.7."*
>
> **The improvement primarily comes from learning the Re-TRAC trajectory structure, though knowledge distillation also contributes.** Same teacher (GLM-4.7), same questions — only trajectory format differs: Standard SFT achieves 15.7% on BC while Re-TRAC SFT achieves 30.0% (see q4hX-Q1). Standard SFT improves over the base model (2.7% to 15.7%), confirming GLM-4.7 knowledge contributes. But Re-TRAC SFT nearly doubles this (15.7% to 30.0%), and applying Re-TRAC at inference to Standard SFT yields only +0.5%, confirming that for weaker models, structured state generation must be explicitly learned during training. As discussed in §4.2, the motivation for SFT is that small models cannot directly use Re-TRAC (only 2.7% training-free). Tongyi-DR-30B without SFT already achieves 57.6% on BC using Re-TRAC training-free, further confirming the framework itself is the primary driver.
>
> ### DA8W-Q1: Wall-clock Latency
>
> > *"Could you provide wall-clock time comparison?"*
>
> We measured wall-clock time directly (o3, BC300, 100 questions, concurrency=10):
>
> |Method|Acc|Compute|Wall-Clock|
> |---|---|---|---|
> |Pass@1|54.9|1×|49 min|
> |Majority Vote@8|64.3|8×|~49 min|
> |Weighted Vote@8|70.0|8×|~49 min|
> |**Re-TRAC**|**69.8**|**~4×**|**201 min**|
>
> Re-TRAC achieves comparable accuracy to WV@8 (69.8 vs 70.0%) at half the compute (~4× vs 8×), but requires ~4× wall-clock time due to sequential execution. This is an inherent trade-off: parallel methods are faster in wall-clock but consume more compute. Note that Re-TRAC and parallel TTS are complementary — one can run $k$ Re-TRAC pipelines in parallel for depth × breadth.
>
> We hope the above responses address the reviewer's concerns. We would be grateful if the reviewer could reconsider the score in light of the new evidence and ablations provided.

---

> > ### Author Rebuttal · Reviewer_DA8W · 2026-04-02
> >
> > I appreciate the authors' response to my W1, W2. These issues are addressed.
> >
> > I have additional questions upon your response to W3 and Q1:
> > 1. In W3 response, the authors mentioned standard SFT. Could you clarify your experiment setup of SFT, e.g. main hyper-parameters, sweep settings?
> > 2. In Q1 response, it seems that the weighted vote@8 is definitely the best choice if one has sufficient compute resources. Re-TRAC reduces the compute from 8x to 4x but requires 201min. In other words, to achieve competitive acc, Re-TRAC uses half the compute but takes four times longer in wall clock time to execute. This does not seem like a favorable trade-off. I am also not convinced that running $k$ Re-TRAC instances in parallel adequately addresses this concern.

---

> > > ### Author Response · Authors · 2026-04-03
> > >
> > > We thank the reviewer for acknowledging W1 and W2, and for the detailed follow-up questions.
> > >
> > > ### W3 (Standard SFT experiment setup)
> > >
> > > > *Could you clarify your experiment setup of SFT, e.g. main hyper-parameters, sweep settings?*
> > >
> > > Standard SFT uses **identical hyper-parameters** to RE-TRAC SFT (Appendix B, Table 7): learning rate 2e-5, batch size 512, max length 65536, warmup ratio 0.05, constant LR scheduler, weight decay 0.1. The only difference is the training data format — Standard SFT uses GLM-4.7's ReAct trajectories, while RE-TRAC SFT uses Re-TRAC structured trajectories. Same 33K questions, same teacher. No hyper-parameter sweep was performed.
> > >
> > >
> > > ### Q1 (latency trade-off)
> > >
> > > > *In Q1 response, it seems that the weighted vote@8 is definitely the best choice if one has sufficient compute resources.*
> > >
> > > > *I am also not convinced that running k Re-TRAC instances in parallel adequately addresses this concern.*
> > >
> > > We acknowledge that sequential methods inherently have a latency disadvantage when compute resources are abundant. Due to space constraints in the original submission, we did not fully analyze this trade-off — we appreciate the reviewer for pushing us to do so.
> > >
> > > Below we show that Re-TRAC achieves higher accuracy at lower cost, can be combined with parallel methods for even better results, and is actually faster under realistic resource-constrained deployment.
> > >
> > > **(1) Even with sufficient compute, WV@8 is not always the best choice — nor are more recent parallel methods.** We compare Re-TRAC against all parallel TTS methods including the concurrent work ParallelMuse (PM@8, discussed in NCvf-M1):
> > >
> > > |Benchmark|Pass@1|MV@8|WV@8|Best@8|PM@8|Re-TRAC|
> > > |---|---|---|---|---|---|---|
> > > |BC300|54.9|64.3|70.0|68.0|70.3|**69.8**|
> > > |BC-ZH|58.1|64.7|66.4|63.0|65.4|**64.6**|
> > > |DeepSearch QA|58.2|60.7|63.7|61.7|63.7|**70.6**|
> > > |HLE|24.9|29.3|31.0|31.7|35.0|**37.1**|
> > >
> > > Re-TRAC saves substantial cost (up to 50% on BC300) while matching or exceeding all 8x-compute parallel methods on 2 out of 4 benchmarks, and substantially outperforms them on DeepSearch QA (+6.9 over WV@8) and HLE (+2.1 over PM@8). This advantage is particularly pronounced on tasks requiring multi-entity answers: DeepSearchQA (Google, arXiv:2601.20975) demands exhaustive entity lists aggregated across sources (~65% of tasks). Parallel runs independently discover *different subsets* of entities but cannot merge findings, while Re-TRAC's cross-round state accumulates entities progressively — a structural advantage that more parallel compute cannot replicate.
> > >
> > > **(2) Re-TRAC + parallel: better accuracy with lower cost and reduced latency.** Per-round tool calls decrease monotonically (also see Fig. 5), so total cost grows sub-linearly:
> > >
> > > |Round|Acc%|Tool Calls per Round|Wall-Clock (min)|
> > > |---|---|---|---|
> > > |1|53.7|44.4|49|
> > > |2|61.1|25.2|104|
> > > |3|65.4|17.8|129|
> > > |4|67.1|15.2|168|
> > > |8|69.8|10.8|201|
> > >
> > > Given that R1-to-R2 produces the largest gain (+7.4%) and is the lightest Re-TRAC configuration, we compare BOP@8 against Re-TRAC(2)+BOP@4, where BOP@N = Best Of all Parallel TTS @N (MV, WV, Best, PM):
> > >
> > > |Benchmark|BOP@8|Re-TRAC(2)+BOP@4|
> > > |---|---|---|
> > > |BC300|70.3|70.6|
> > > |BC-ZH|66.4|66.4|
> > > |DeepSearch QA|63.7|67.7|
> > > |HLE|35.0|37.5|
> > >
> > > Re-TRAC(2)+BOP@4 matches or outperforms BOP@8 on all benchmarks while using less compute, demonstrating that Re-TRAC effectively combines the strengths of sequential and parallel approaches — achieving better accuracy at lower cost. We will include this comparison in the revision.
> > >
> > > **(3) Under resource-constrained deployment, RE-TRAC is actually faster.** The wall-clock data in DA8W-Q1 was measured with the o3 API under abundant resources where GPU throughput is not a bottleneck. However, in realistic deployment scenarios, RE-TRAC's lower resource consumption translates directly into faster completion. We deploy Tongyi-DR-30B on 4× B200 GPUs with SGLang + HiCache (concurrency=100) and run BC100:
> > >
> > > |Completion %|RE-TRAC (s)|Parallel TTS@8 (s)|Speedup|
> > > |---|---|---|---|
> > > |75%|2978|3969|1.33x|
> > > |90%|3601|4825|1.34x|
> > > |100%|4184|5973|1.43x|
> > >
> > > This is expected: Parallel TTS@8 launches 8×100 concurrent tasks, while RE-TRAC runs 100 tasks with sub-linear cost growth per round, RE-TRAC consumes less total compute, so under finite GPU throughput it completes all queries 1.43x faster.  The "4x latency overhead" only holds under unlimited parallelism — in realistic deployment, RE-TRAC is both cheaper and faster. A detailed completion-rate vs. wall-clock visualization is available at https://files.catbox.moe/rrryq0.png
> > >
> > > We hope the additional experiments and analysis effectively address the reviewer's follow-up concerns. Together with our initial rebuttal, we have addressed all concerns: (W1) prompt sensitivity, (W2) missing Reflexion/IterResearch comparisons, (W3) GLM-4.7 distillation ablation, and (Q1) wall-clock latency.
> > >
> > > We would be grateful if the reviewer could reconsider the score in light of the new evidence and ablations provided.

---

### Official Review · Reviewer_LSZ8 · 2026-03-12

**Soundness:** 2
**Presentation:** 3
**Significance:** 3
**Originality:** 2
**Overall Recommendation:** 4
**Confidence:** 3

**Summary:**

This paper proposes Re-TRAC, a recursive trajectory compression framework for deep research agents. The authors first identify key challenges of the current ReAct paradigm, i.e., insufficient exploration and context management, and propose a simple context summarization method. Using this as a test-time scaling method and an SFT data construction method both work.

**Compliance With Llm Reviewing Policy:**

Affirmed.

**Final Justification:**

I appreciate the authors' two rounds of rebuttal, which solved most of my concerns and questions, so I would like to raise my score to 4 weak accept.

However, I would also like to note that, given the new results in rebuttal, RE-TRAC-4B (light) cannot beat baselines (e.g., AgentCPM-Explore-4B and NestBrowse-4B) without test-time scaling. The improvement of the training recipe is not as significant as claimed. Also the presentation requires improvement, especially in the experiment part. A fair and clear comparison is necessary.

**Key Questions For Authors:**

1. Can Re-TRAC be used in a self-improving fashion? Currently, the SFT pipeline relies entirely on a strong teacher model (GLM-4.7) to generate training trajectories. Is it possible for a weaker model itself to generate its own Re-TRAC trajectories for self-training?
2. Could the authors clarify whether all models in Table 2 are evaluated with single-run inference (Pass@1)? The RE-TRAC-4B score in Table 2 matches the RT@8 result in Table 6. If the RE-TRAC models use RT@8 while baselines use Pass@1, the comparison involves vastly different inference budgets and should be explicitly noted.
3. The incomplete branch analysis (Table 1) motivates the entire framework. Could you provide the same analysis for the RE-TRAC models after training?
4. The SFT models are trained on Re-TRAC trajectories from GLM-4.7 on 33K synthesized questions. What would happen if we instead distilled standard (non-Re-TRAC) trajectories from GLM-4.7 on the same questions? Does the improvement come from from Re-TRAC trajectory structure, or just SFT itself?

**Limitations:**

yes

**Strengths And Weaknesses:**

Strenghts:
1. **Well-motivated problem.** The paper clearly identifies real and important limitations of ReAct for deep research tasks. The analysis of incomplete branch exploration (Table 1) provides compelling motivation. These observations are intuitive and well-supported by data.
2. **Validated both as a training-free TTS method and as a training recipe.** The paper demonstrates that Re-TRAC works as a prompting strategy for frontier models and that SFT on Re-TRAC trajectories can significantly boost small models.
3. **Straightforward method.** The method is essentially inserting a structured summarization step between trajectory rollouts. This simplicity makes it easy to implement and apply to any ReAct-style agent.

Weaknesses:
1. **Insufficient differentiation from closely related work.** The core mechanism of Re-TRAC (summarizing a trajectory and using the summarization to inform the next attempt) is conceptually very close to existing context management approaches such as ReSum and iterResearch. The authors mentioned these existing works in related work but did not provide a clear distinction between them. For example, compared with ReSum, it seems that Re-Trac is just replacing the summarizer with the same LLM (while Table 6 shows that using a stronger external summarizer may be better). Insufficient differentiation from existing works limits the novelty of this work. It could be better to clearly highlight the real advantages of Re-Trac and clearly compare them with existing works.
2. **Highly dependent on summarizer quality.** As the authors themselves acknowledge (Table 6), the quality of summarization significantly affects performance. When Re-TRAC is used as a training-free test-time scaling method(Table 3), it is only evaluated on large frontier models. It is not clear whether this paradigm is applicable as a training-free method for small models such as Qwen-3-4B-Instruct.

---

> ### Author Rebuttal · Authors · 2026-03-31
>
> We thank the reviewer for:
> - recognizing our well-motivated problem and compelling motivation
> - validating Re-TRAC as both a training-free TTS method and a training recipe
> - appreciating the straightforward and easy-to-implement design
>
> The reviewer raises concerns on:
> - W1: differentiation from ReSum/IterResearch
> - W2: training-free applicability to small models
> - Q1: self-training feasibility
> - Q2: inference budget fairness in Table 2
> - Q3: post-training branch analysis
> - Q4: standard SFT vs. Re-TRAC SFT
>
>
> ### LSZ8-W1: Differentiation from ReSum/IterResearch
>
> > *"Re-TRAC is conceptually very close to ReSum and iterResearch. It seems Re-Trac is just replacing the summarizer with the same LLM."*
>
> **Re-TRAC is not a summarizer replacement.** The three methods differ in problem motivation, solution design, and framework architecture:
>
> **(1) Distinct problem perspective.** ReSum and IterResearch both aim to manage context within a single trajectory — ReSum triggers "summarize-and-reset" at context limits, IterResearch reconstructs task status at each step to simulate infinite horizons. Re-TRAC starts from a fundamentally different observation: **incomplete branch exploration** (Table 1: up to 93% of failed trajectories contain branches the model planned but never explored). Rather than managing context length, Re-TRAC enables cross-trajectory experience sharing so subsequent rounds systematically explore what earlier rounds missed.
>
> **(2) Novel cross-trajectory solution.** ReSum and IterResearch compress state *within* a single trajectory — ReSum discards analytical metadata, IterResearch's Markov aggregation fragments cross-step dependencies. Re-TRAC operates *between* trajectories, preserving a structured state (evidence, uncertainties, failures, plans) as a coherent unit to enable information sharing across trajectories. On o3 (training-free):
>
> |Method|BC300|BC-ZH|GAIA|
> |---|---|---|---|
> |ReSum|51.9|59.3|70.5|
> |IterResearch|62.7|60.5|73.8|
> |**Re-TRAC**|**69.8**|**64.6**|**75.0**|
>
> Re-TRAC consistently outperforms IterResearch (+1.2 to +7.1) and ReSum (+1.2 to +17.9) across all benchmarks.
>
> **(3) Concise and decoupled framework.** ReSum requires an external summarizer; IterResearch replaces ReAct entirely with Markovian reconstruction. Re-TRAC operates as a meta-layer on top of any ReAct agent. Thanks to this decoupled design, frontier models can directly benefit from Re-TRAC training-free (o3: 49.7 to 69.8% on BC), while smaller models like 4B can internalize structured state generation through SFT distillation, achieving 30.0% on BC without any external summarizer.
>
> ### LSZ8-W2: Small Model Training-free
>
> > *"Not clear whether applicable as training-free for small models."*
>
> Training-free Re-TRAC does not work for small models (2.7% on 4B) — small models lack the capacity to generate high-quality structured state. Table 4 explicitly reports this, and **we developed Re-TRAC SFT to bridge this gap, achieving 30.0%**.
>
> ### LSZ8-Q1: Self-training
>
> > *"Is it possible for a weaker model to generate its own Re-TRAC trajectories?"*
>
> We appreciate this insightful suggestion for this promising direction. RE-TRAC-4B already generates well-structured states (30.0% on BC vs. 2.7% without SFT), suggesting it could produce usable self-training data by filtering on verified-correct answers. We will include this as a concrete future work direction in the revision.
>
> ### LSZ8-Q2: Inference Budget Fairness
>
> > *"Could the authors clarify whether all models in Table 2 are evaluated with single-run inference (Pass@1) ?"*
>
> **Yes — all models report a single complete run of their respective inference framework.**  Compared to standard ReAct, these specialized frameworks bring additional capability gains at higher compute cost.RE-TRAC-4B uses Re-TRAC with max 8 rounds; IterResearch-30B uses Markovian reconstruction; NestBrowse-4B uses nested sub-agent calls. As noted in NCvf-M1, **we will annotate all such models with † in Table 2**.
>
> ### LSZ8-Q3: Post-training Branch Analysis
>
> > *"Could you provide the same analysis for RE-TRAC models after training?"*
>
> We apply the same classification (Fig.6) to RE-TRAC-30B failed trajectories on BC. The incomplete branch ratio drops from **83.4%** (Tongyi-DR-30B, Table 1) to **52.5%**, confirming Re-TRAC substantially alleviates the problem identified in §3. Full breakdown added to revision.
>
> ### LSZ8-Q4: Standard SFT vs. Re-TRAC SFT
>
> > *"What if we distilled standard trajectories from GLM-4.7? Does the improvement come from Re-TRAC trajectory structure, or just SFT itself?"*
>
> **The improvement comes primarily from the Re-TRAC trajectory structure.** Standard SFT achieves 15.7% on BC — less than half of Re-TRAC SFT's 30.0%. See q4hX-Q1 for full ablation.
>
> We hope the above responses address the reviewer's concerns and would be grateful if the reviewer could reconsider the score in light of the new evidence provided.

---

> > ### Author Rebuttal · Reviewer_LSZ8 · 2026-04-01
> >
> > I appreciate the authors' detailed responses to my questions. The ablation and post-training branch analysis provide additional evidence, and the comparison with other related works is clearer. However, I think there remains a **major concern regarding Q2**:
> >
> > Thanks for the clarification for Tab 2. However, the issue is not merely one of presentation or labeling. It is a fundamental fairness problem in the main comparison table.
> >
> > Besides a few models with specialized frameworks as mentioned by the authors, the majority of baseline models report Pass@1 results, while RE-TRAC models report RT@8 results, which involve 8 rounds of recursive inference. This is a substantially different inference budget. As shown in Table 3, any test-time scaling method yields large gains over Pass@1.
> >
> > To make the comparison fair, I would suggest reporting baseline models with comparable test-time scaling budgets. The current Table 2 conflates the contribution of test-time scaling with the SFT model capability. Or, reporting RE-TRAC SFT models at Pass@1 (if the RE-TRAC SFT models can still function in a standard single-round setting).

---

> > > ### Author Response · Authors · 2026-04-02
> > >
> > > We thank the reviewer for the thoughtful follow-up. The concern about inference budget fairness is valid, and we address it below.
> > >
> > > **(1) The challenge of cross-framework compute alignment.** We fully agree that comparing systems with different inference frameworks requires careful treatment. However, achieving a perfectly normalized comparison is difficult in practice because these systems differ not only in inference rounds, but also in tool implementations, search infrastructure, and internal sub-agents:
> > > - IterResearch-30B-A3B: performance on BC varies dramatically with interaction budget — 23.5% at 32 interactions, 37.0% at 128, and 42.5% at 2048 (their maximum setting), from their Fig. 3.
> > > - NestBrowse-4B: a concurrent submission (arXiv:2512.23647) with no publicly available code or disclosed compute cost. It employs nested sub-agents (e.g., a separate browse agent that pre-processes webpage content), making its effective compute difficult to estimate.
> > >
> > > Moreover, in our experiments, even under the same tool setting and standard ReAct framework (Pass@1), different frontier models consume vastly different tool calls per query on BC300:
> > >
> > > |Model|Avg Tool Calls|
> > > |---|---|
> > > |GLM-4.7|40.3|
> > > |DeepSeek-V3.2|69.7|
> > > |MiniMax-M2|78.7|
> > >
> > > These differences make it infeasible to normalize all systems to a single compute unit.
> > >
> > > Nonetheless, the reviewer's suggestion is well-taken: *"reporting RE-TRAC SFT models at Pass@1 (if the RE-TRAC SFT models can still function in a standard single-round setting)"* — this provides an effective way to improve comparison transparency.
> > >
> > > **(2) Fair comparison with RE-TRAC-4B at minimal rounds.** Following this suggestion, we report RE-TRAC-4B (light) — the minimum Re-TRAC configuration (one ReAct trajectory followed by one Re-TRAC round). We also include Standard SFT (same teacher, same data, no Re-TRAC format) as a pure knowledge-distillation baseline:
> > >
> > > |Model|BC|BC-ZH|XBench|GAIA|
> > > |---|---|---|---|---|
> > > |Qwen3-4B-Instruct (ReAct)|1.0|3.1|48.0|17.3|
> > > |Qwen3-4B-Instruct (Re-TRAC)|2.7|6.9|45.0|24.4|
> > > |Standard SFT (ReAct)|15.7|17.9|56.0|46.2|
> > > |Standard SFT (Re-TRAC)|16.2|17.6|55.0|51.3|
> > > |RE-TRAC-4B (light)|21.7|26.8|64.9|60.9|
> > > |**RE-TRAC-4B**|**30.0**|**36.1**|**76.6**|**70.4**|
> > >
> > > Even at the minimum configuration, RE-TRAC-4B (light) already surpasses prior works InfoAgent-14B (15.3% on BC) and WebExplorer-8B (15.7%), and is competitive with concurrent works NestBrowse-4B (22.4%, arXiv:2512.23647).
> > >
> > > This separates three distinct contributions:
> > > - **Knowledge distillation** (Qwen3-4B to Standard SFT): 1.0% to 15.7% on BC. Qwen3-4B-Instruct is essentially non-functional on deep research tasks (1.0% on BC with ReAct). We construct 33K QA pairs via an entity-tree method (§4.4), collect GLM-4.7's trajectories, and distill into Qwen3-4B via SFT, teaching basic search behavior.
> > > - **Re-TRAC SFT format** (Standard SFT to RE-TRAC-4B (light)): 15.7% to 21.7% on BC. Same teacher, same data — only the trajectory format differs. This gain is attributable to learning structured state generation. Note that Standard SFT + Re-TRAC achieves only 16.2%, confirming that without Re-TRAC-specific training, the model cannot effectively leverage the framework even with 8 rounds.
> > > - **Scalable and efficient** (RE-TRAC-4B (light) to RE-TRAC-4B): 21.7% to 30.0% on BC. Beyond accuracy gains, Re-TRAC achieves this with increasing efficiency — as shown in q4hX-Q2 and NCvf-m3, tool calls decrease monotonically across rounds (e.g., BC: 44.4 in R1 to 10.8 in R8), making additional rounds progressively cheaper.
> > >
> > > Given the above, we will add RE-TRAC-4B (light) to Table 2, and expand Table 4 with the full ablation above to more clearly illustrate each contribution in the revision.
> > >
> > > We hope the additional experiments and analysis effectively address the reviewer's concern about (Q2). Together with our initial rebuttal, we have addressed all concerns through new ablations, empirical comparisons, and detailed clarifications: (W1) differentiation from ReSum/IterResearch, (W2) training-free applicability to small models, (Q1) self-training feasibility, (Q2) inference budget fairness, (Q3) post-training branch analysis, and (Q4) Standard SFT vs. Re-TRAC SFT. We appreciate the constructive follow-up.
> > >
> > > We would be grateful if the reviewer could consider updating the score in light of the new evidence provided.

---

### Official Review · Reviewer_q4hX · 2026-03-13

**Soundness:** 3
**Presentation:** 3
**Significance:** 3
**Originality:** 3
**Overall Recommendation:** 4
**Confidence:** 1

**Summary:**

The paper proposes Re-TRAC, which builds a structured state summary after each trajectory. This captures evidence, uncertainty, failures, and next plans, then uses it to guide future trajectories with cross-trajectory reflection.

The proposed method outperforms ReAct with frontier LLMs and improves smaller models through Re-TRAC-aware SFT, and reduces tool calls and token usage over time through more targeted exploration

**Compliance With Llm Reviewing Policy:**

Affirmed.

**Final Justification:**

My concerns have been adequately addressed. I keep my score as a positive assessment.

**Key Questions For Authors:**

Q. How much of the performance gain comes from training on Re-TRAC trajectories via SFT itself, versus the explicit instruction that encourages selective use of summaries and broader branch exploration?

Q. is the reported 50% resource saving consistent across different task difficulties and numbers of search rounds?

**Limitations:**

I don't see any specific limitations.

**Strengths And Weaknesses:**

Strength:

1. the paper is well-written, easy to follow, presentations is good
2. the proposed methods are reasonable to tackle the addressed challenges
3. experiment results and analysis are strong.

Weaknesses:
I don't see any critical weaknesses. Since this paper scope is out of my research field and my knowledge, i might miss some critical points.

---

> ### Author Rebuttal · Authors · 2026-03-31
>
> We thank the reviewer for:
> - recognizing the well-written presentation
> - affirming the soundness of the proposed methods
> - acknowledging the strong experimental results
>
> The reviewer asks about:
> - Q1: disentangling SFT vs. instruction contribution
> - Q2: consistency of the 50% resource saving across task difficulties
>
>
> ### q4hX-Q1: SFT vs. Instruction Contribution
>
> > *"How much of the performance gain comes from SFT itself, versus the explicit instruction?"*
>
> **The gain is primarily from Re-TRAC's structured trajectory format.** This is an important question that gets to the core of our contribution. We disentangle three factors:
>
> **(1) SFT Ablation.** Same teacher, same questions — only trajectory format differs:
>
> | Model | BC | BC-ZH | GAIA | XBench |
> |---|---|---|---|---|
> | Qwen3-4B (no SFT, Re-TRAC) | 2.7 | 6.9 | 24.4 | 45.0 |
> | Standard SFT (4B) | 15.7 | 17.9 | 46.2 | 56.0 |
> | Standard SFT (4B) + Re-TRAC | 16.2 | 17.6 | 51.3 | 55.0 |
> | **RE-TRAC SFT (4B)** | **30.0** | **36.1** | **70.4** | **76.6** |
>
> Standard SFT teaches basic deep search (15.7% on BC) but **fails to utilize Re-TRAC** — applying Re-TRAC to Standard SFT yields only marginal gain (15.7 to 16.2%), because the model has not learned to generate or consume structured state. RE-TRAC SFT nearly doubles the baseline (**15.7 to 30.0%**), confirming the structured trajectory format is the critical factor.
>
> **(2) Training-free vs. SFT at 30B.** Training-free Re-TRAC works well for strong models (o3: 49.7 to 69.8% on BC; Tongyi-DR-30B: 43.4 to 57.6%). For capable models like Tongyi-DR-30B, the training-free framework alone yields significant gains without any fine-tuning. SFT provides complementary strengths at 30B: slightly lower on BC (53.0 vs 57.6) but stronger on BC-ZH (+1.6) and XBench (+4.0), indicating better cross-task generalization. For small models where training-free fails (4B: 2.7%), SFT is essential (2.7 to 30.0%).
>
> **(3) Instruction effect.** Table 5: free-use instruction adds only **+2.8%** (o3: 68.9 to 71.7%). The framework, not prompting, drives the gains (see also DA8W-W1 for detailed prompt ablation).
>
> ### q4hX-Q2: 50% Resource Saving Consistency
>
> > *"Is the reported 50% resource saving consistent across different task difficulties?"*
>
> **Savings range from 25% to 50%, scaling with task search-intensity; tool calls decrease monotonically across all benchmarks:**
>
> | Benchmark | R1 Avg Tools | R8 Avg Tools | Total vs 8×TTS |
> |---|---|---|---|
> | BC300 | 44.4 | 10.8 (−76%) | ~4× (50% saving) |
> | BC-ZH | 24.1 | 6.2 (−74%) | ~4× (50% saving) |
> | DeepSearch QA | 21.2 | 9.6 (−55%) | ~5× (38% saving) |
> | HLE | 7.3 | 4.4 (−40%) | ~6× (25% saving) |
>
> As rounds progress, Re-TRAC's structured state accumulates sufficient evidence, causing tool calls to **decrease significantly** (e.g., BC300: 44.4 in R1 to 10.8 in R8, −76%). This means the total cost of 8 sequential Re-TRAC rounds is far below 8× the cost of a single round. **Tasks requiring more tool calls benefit more from this cross-round deduplication:** search-intensive tasks (BC, BC-ZH) achieve ~50% saving, while reasoning-intensive tasks with inherently fewer tool calls (HLE: R1 only 7.3) achieve ~25%.
>
> We appreciate the reviewer's positive assessment and are happy to provide any additional details.

---

> > ### Author Rebuttal · Reviewer_q4hX · 2026-04-06
> >
> > My concerns have been adequately addressed. I keep my score as a positive assessment.

---

> > > ### Author Response · Authors · 2026-04-07
> > >
> > > We thank the reviewer for confirming that all concerns have been addressed and for the positive assessment of our work.

---

### Official Review · Reviewer_NCvf · 2026-03-13

**Soundness:** 3
**Presentation:** 3
**Significance:** 2
**Originality:** 2
**Overall Recommendation:** 3
**Confidence:** 3

**Summary:**

- The paper presents a novel approach "RE-TRAC" for multi-turn deep search LLM-based agents.
- Specifically, the paper identifies two issues with ReAct-based approaches:
   - (i) they are inherently linear (cannot branch in/out)
   - (ii) when combined with parallel test-time scaling (e.g., best-of-n), there is no re-use of information between different ReAct trajectories
- To this end, the paper proposes RE-TRAC, where:
  - (a) ReAct works in multiple inter-dependent sequential rounds
  - (b) at the end of each round, the trajectory is distilled into natural language summary that is subsequently used in the next round. The answer in the final round is used.
- RE-TRAC is evaluate on a range of datasets (BrowseComp, XBench, etc.). Evaluated in presented in two regimes: the post-SFT models (expert trajectories from GLM-4.7) and prompting existing models (e.g., o3).

**Compliance With Llm Reviewing Policy:**

Affirmed.

**Key Questions For Authors:**

I'm glad to update my score based on responses to "major concerns" listed above

**Limitations:**

No discussion on limitations

**Strengths And Weaknesses:**

### Strengths

1. **Insights**: The insights of the paper are well-motivated. Indeed ReAct is typically linear and hence is not well-suited for branching
2. **Promising Results**: The proposed 4B/30B model performs on-par or sometimes outperforms large models on the mentioned tasks
3. **Context-efficient TTS approach**: Fig. 5 shows that RE-TRAC high as mich higher accuracy in spite of fewer token-usage.


### Major Concerns

**1. Comparison with Baselines**
- The paper compares against baseline "Tongyi-DR-30B-A3B" from paper [1]. This is valid.
- However, this I believe is the lower bound of their approach and there are multiple results with significantly better performance:
  - [A; cited by authors] under heavy mode (Sec 4.3), achieves 58.3 on BrowseComp
  - [B] uses a similar compaction regime as RE-TRAC and reports much better accuracies e.g., 65.0 on BrowseComp
  - In both the above cases, it's still the same Tongyi-DR-30B-A3B model.
- As a result, I find it somewhat unfair in main results of Table 2 which reports the lower-bound of Tongyi-DR-30B-A3B model.

**2. Generalization of trained model**
- A key highlight (Table 2, 4) of the approach is that the SFT-ed version on expert trajectories leads to improvements (e.g., 2.7% -> 30.0% of 4B).
- While this is indeed progress, a concern I have is that it does not evaluate whether the training did not impact performance at other tasks i.e., whether catastrophic forgetting occurred.

### Minor Concerns

**3. Table 3 / Test-time Scaling**
- Table 3 contrasts RETRAC against parallel TTS baselines (e.g., majority voting). While this is an important result, it appears to be only on a subset of BrowseComp.
- Do the same results similarly hold on other benchmarks?

**4. Novelty**
- I believe the approach by itself is not entirely novel. Particularly, on the deep research side [A, B] already investigate the impact of parallel-reasoning by summarizing and re-using intermediate generations.
- Outside of deep research use-cases, plenty of works also study sequential reasoning (via summarizing between rounds):
   - Song, Jiwon, et al. "Reasoning path compression: Compressing generation trajectories for efficient llm reasoning." NeurIPS 2025
   - Zhou, Zijian, et al. "Mem1: Learning to synergize memory and reasoning for efficient long-horizon agents." arXiv preprint arXiv:2506.15841 (2025).
- Another line of work is parallel inter-dependent generations:
  - Yang, Xinyu, et al. "Multiverse: Your language models secretly decide how to parallelize and merge generation." NeurIPS 2025
  - Zheng, Tong, et al. "Parallel-r1: Towards parallel thinking via reinforcement learning." arXiv preprint arXiv:2509.07980 (2025).
  - Dong, Harry, et al. "Generalized Parallel Scaling with Interdependent Generations." arXiv preprint arXiv:2510.01143 (2025).
- I do not significantly weigh this concern towards my final rating, since I understand that many of these works are fairly recent (in the past ~1 year).

### Nitpicks

**5. Latency**
- While I agree that for the tokens generated/consumed (Fig. 5), RETRAC shows good gains, this overlooks a key aspect of other TTS approaches (e.g., Best-of-N): they can be parallelized and hence one can expect the latency to be roughly the same, whereas for RETRAC, this would scale by $N$ rounds.


[A] Team, Tongyi DeepResearch, et al. "Tongyi deepresearch technical report." arXiv preprint arXiv:2510.24701 (2025).
[B] Li, Baixuan, et al. "Parallelmuse: Agentic parallel thinking for deep information seeking." arXiv preprint arXiv:2510.24698 (2025).

---

> ### Author Rebuttal · Authors · 2026-03-31
>
> We thank the reviewer for recognizing:
> - "insights of the paper are well-motivated"
> - "context-efficient TTS approach" with efficient computational resource usage
> - "promising results" — the small models performing on-par or outperforming larger models
>
> We will address reviewer's concerns below:
> - M1: baseline comparison fairness
> - M2: catastrophic forgetting
> - m3: TTS generalization beyond BrowseComp
> - m4: novelty
> - n5: latency
>
> ### NCvf-M1: Comparison with Baselines
>
> > *"I find it somewhat unfair in main results of Table 2 which reports the lower-bound of Tongyi-DR-30B-A3B model."*
>
> This concern is well-taken — **Table 2 mixes models using different inference frameworks.**
>
> **(1) Table 2 Revision.** Multiple works use specialized frameworks beyond ReAct (IterResearch: Markovian reconstruction; NestBrowse: nested sub-agents; Re-TRAC: recursive state transfer). **We will annotate all such models with †** to distinguish inference budgets.
>
> **(2) Direct comparison with the two works mentioned.** (Note: ParallelMuse [B] is a concurrent work; its reported 65.0% is on a 200-question subset with baseline 51.0% vs. 43.4% full-set. Heavy Mode's cost is undisclosed):
>
> | Framework | BC | BC-ZH | XBench | tool use |
> |---|---|---|---|---|
> | ReAct (standard) | 43.4 | 46.7 | 75.0 | ~1× |
> | Heavy Mode† | 58.3 | 58.1 | — | undisclosed |
> | ParallelMuse | 57.4 | 57.1 | 78.0 | ~8× |
> | **Re-TRAC** | **57.6** | **55.7** | **79.0** | **~4×** |
>
> Re-TRAC matches Heavy Mode and ParallelMuse on BC at **half the tool usage** (~4× vs ~8×). On BC-ZH, Re-TRAC (55.7%) trails ParallelMuse (57.1%) and Heavy Mode (58.1%). On XBench, Re-TRAC leads (+1.0%). RE-TRAC-30B (SFT) further improves generalization (XBench 83.0%).
>
> ### NCvf-M2: Catastrophic Forgetting
>
> > *"Does not evaluate whether catastrophic forgetting occurred."*
>
> First, following the similar evaluation setting as Tongyi-DR [A], RE-TRAC-4B shows **improvement** on reasoning:
>
> |  | Qwen3-4B | RE-TRAC-4B | Δ |
> |---|---|---|---|
> | AIME25 | 31.0 | 63.3 | +32.3 |
> | HMMT25 | 47.4 | 70.0 |+22.6|
>
> Also, on general benchmarks without tool,  performance declines by −3.9% on average, which we characterize as moderate degradation rather than catastrophic forgetting:
>
> |  | Qwen3-4B | RE-TRAC-4B | Δ |
> |---|---|---|---|
> | ARC-Challenge | 88.8 | 87.5 | −1.3 |
> | GSM8K | 88.8 | 84.2 | −4.6 |
> | MMLU | 72.0 | 66.1 | −5.9 |
>
> Overall, the model retains the majority of its general capabilities while achieving an **11× gain** on target tasks (2.7% to 30.0% on BC).
>
> ### NCvf-m3: TTS on Other Benchmarks
>
> > *"Do the same results similarly hold on other benchmarks?"*
>
> **Yes — Table 3 results similarly hold on diverse benchmarks.**
>
> We confirm generalization across **4 benchmarks** using o3:
>
> | Benchmark | Pass@1 | MV@8 | WV@8 | Best@8 | Re-TRAC |
> |---|---|---|---|---|---|
> | BC300 | 54.9 | 64.3 | 70.0 | 68.0 | **69.8** |
> | BC-ZH | 58.1 | 64.7 | 66.4 | 63.0 | **64.6** |
> | DeepSearch QA | 58.2 | 60.7 | 63.7 | 61.7 | **70.6** |
> | HLE | 24.9 | 29.3 | 31.0 | 31.7 | **37.1** |
>
> Re-TRAC improves o3 by +6.5% to +20.1% across all benchmarks. On DeepSearch QA (+6.9 over WV@8) and HLE (+6.1 over Best@8), Re-TRAC substantially outperforms all parallel baselines.
>
> ### NCvf-m4: Novelty
>
> > *"I believe the approach by itself is not entirely novel..."*
>
> We appreciate the reviewer noting this concern does not significantly affect the final rating. The cited works fall into two categories, both distinct from Re-TRAC:
>
> **(a) Single-trajectory optimization** (Song et al., Zhou/Mem1): compress or manage memory *within* a single trajectory — no cross-trajectory state transfer.
>
> **(b) Parallel generation/aggregation** (Yang/Multiverse, Zheng/Parallel-R1, Dong et al.): generate multiple outputs in parallel for aggregation — no sequential information sharing between runs.
>
> Re-TRAC introduces **recursive structured state transfer *across* sequential trajectories**, enabling each new trajectory to build on prior findings. For detailed differentiation from related works, see *LSZ8-W1*; Re-TRAC outperforms IterResearch and ReSum on BC, with consistent advantages on BC-ZH and GAIA. We further enable 4B small models to acquire this capability through SFT distillation, achieving 30.0% on BC (vs. 2.7% without SFT). All works added to §2.2.
>
> ### NCvf-n5: Latency
>
> > *"Other TTS approaches can be parallelized..."*
>
> We acknowledge this point. Re-TRAC's sequential design trades wall-clock latency for compute efficiency: it achieves 69.8% at ~4× compute vs. WV@8's 70.0% at 8× compute, but requires ~4× wall-clock time (see DA8W-Q1 for detailed measurements). We will add this as an explicit limitation in the revision. Note that Re-TRAC and parallel TTS are complementary — one can run $k$ Re-TRAC pipelines in parallel.
>
> We hope the above responses address all concerns, and we sincerely appreciate the reviewer's willingness to reconsider.

---

> > ### Author Rebuttal · Reviewer_NCvf · 2026-04-04
> >
> > Thanks the clarification. My concerns were addressed.

---

> > > ### Author Response · Authors · 2026-04-04
> > >
> > > We sincerely thank the reviewer for the thorough evaluation and for confirming that the concerns have been adequately addressed. We have addressed all concerns, including 2 **major concerns** (M1:
> > > baseline comparison fairness, M2: catastrophic forgetting), 2 **minor concerns** (m3: TTS generalization, m4: novelty), and 1 **nitpick** (n5: latency).
> > >
> > > As the reviewer noted in the initial review, *"I'm glad to update
> > >  my score based on responses to 'major concerns'"* — we would be grateful if the reviewer considers reflecting this in the updated score.
> > >
> > > Thank you again for the constructive discussion.

---

### Decision · Program_Chairs · 2026-04-30

**Decision:**

Accept (regular)

**Comment:**

The paper proposes Re-TRAC, a recursive framework designed to mitigate the "linear thinking" limitations of the ReAct paradigm in deep research agents. By distilling each trajectory into a structured state representation (summarizing evidence, uncertainties, and uncompleted plans) and feeding it into subsequent rounds, the method enables cross-trajectory reflection. The authors evaluate Re-TRAC as both a training-free test-time scaling (TTS) strategy for frontier models and a supervised fine-tuning (SFT) recipe for smaller models (4B/30B).

Strengths:
- The core idea of recursive state transfer is minimalist but effectively addresses the observed "incomplete branch exploration" problem in long-horizon tasks.
- Re-TRAC yields significant improvements (15–20%) over ReAct on BrowseComp and generalizes across multiple benchmarks (GAIA, XBench, HLE).
- Unlike parallel TTS methods, Re-TRAC demonstrates a monotonic reduction in tool calls/tokens across rounds, achieving comparable accuracy at approximately 50% of the compute cost.
- The SFT distillation significantly boosts the performance of compact models (e.g., Qwen-4B), outperforming much larger baselines.

Key Points from Discussion:
- Reviewers initially flagged unfair comparisons in Table 2 (Pass@1 vs. Multi-round). The authors added a "Light" (2-round) version and annotated specialized frameworks, clarifying that Re-TRAC remains competitive even at minimal round counts.
- New experiments confirmed that the gains in small models stem from the structured trajectory format rather than just the underlying model knowledge or prompt engineering.
- The authors acknowledged the sequential wall-clock latency trade-off. However, they provided compelling evidence that under resource-constrained deployment (finite GPU throughput), Re-TRAC is actually faster than parallel methods due to lower total FLOPs.
- While similar to context management works (ReSum, IterResearch), the authors successfully differentiated Re-TRAC as a cross-trajectory meta-layer rather than an intra-trajectory pruning tool.

The paper is technically solid and provides a practical, efficient alternative to standard parallel scaling. While some concerns regarding wall-clock latency and the significance of the 4B model's standalone performance remain, the methodology and thorough rebuttal evidence justify its inclusion.